EMBO
Molecular Medicine

# Therapeutic gene editing in CD34+ hematopoietic progenitors from Fanconi anemia patients

Begoña Diez[1,2,3], Pietro Genovese[4], Francisco J Roman-Rodriguez[1,2,3], Lara Alvarez[1,2,3], Giulia Schiroli[4,5], Laura Ugalde[1,2,3], Sandra Rodriguez-Perales[6], Julian Sevilla[3,7,8], Cristina Diaz de Heredia[9], Michael C Holmes[10], Angelo Lombardo[4,5], Luigi Naldini[4,5] (ID), Juan Antonio Bueren[1,2,3,*] (ID) & Paula Rio[1,2,3,**] (ID)

## Abstract

Gene targeting constitutes a new step in the development of gene therapy for inherited diseases. Although previous studies have shown the feasibility of editing fibroblasts from Fanconi anemia (FA) patients, here we aimed at conducting therapeutic gene editing in clinically relevant cells, such as hematopoietic stem cells (HSCs). In our first experiments, we showed that zinc finger nuclease (ZFN)-mediated insertion of a non-therapeutic EGFP-reporter donor in the *AAVS1* "safe harbor" locus of FA-A lymphoblastic cell lines (LCLs), indicating that FANCA is not essential for the editing of human cells. When the same approach was conducted with therapeutic *FANCA* donors, an efficient phenotypic correction of FA-A LCLs was obtained. Using primary cord blood CD34+ cells from healthy donors, gene targeting was confirmed not only in *in vitro* cultured cells, but also in hematopoietic precursors responsible for the repopulation of primary and secondary immunodeficient mice. Moreover, when similar experiments were conducted with mobilized peripheral blood CD34+ cells from FA-A patients, we could demonstrate for the first time that gene targeting in primary hematopoietic precursors from FA patients is feasible and compatible with the phenotypic correction of these clinically relevant cells.

**Keywords** CD34+ cells; Fanconi anemia; gene editing; hematopoietic stem and progenitor cells; zinc finger nucleases

**Subject Categories** Genetics, Gene Therapy & Genetic Disease; Haematology

## Introduction

Gene editing has recently emerged as a realistic approach for the treatment of both inherited and acquired diseases (Maeder & Gersbach, 2016). Although studies conducted in different cell types, such as fibroblasts or embryonic stem cells, have shown relatively high efficiencies of homology directed repair (HDR) (Reviewed in Urnov et al, 2010), gene editing in hematopoietic stem and progenitor cells (HSPCs) has presented particular difficulties, as compared to other target cells (Lombardo et al, 2007). Significantly, in 2014 Genovese and colleagues (Genovese et al, 2014) showed significant improvements in the gene editing of human long-term hematopoietic SCID-repopulating cells by tailoring delivery platforms and culture conditions.

In spite of advances in the editing of adult HSPCs (Genovese et al, 2014; Hoban et al, 2015; De Ravin et al, 2016), gene-targeting efficiencies currently obtained in these rare precursor cells are still far from transduction rates already achieved with gamma-retroviral and lentiviral vectors (Cartier et al, 2009; Cavazzana-Calvo et al, 2010; Aiuti et al, 2013; Biffi et al, 2013; Hacein-Bey-Abina et al, 2014) Consequently, clinical applications associated with the positive selection of gene-edited cells are particularly attractive. In this respect, either the specific knockout of the *CCR5* receptor of HSPCs from HIV-infected patients (Tebas et al, 2014) or the specific restoration of the *IL2RG* in HSPCs from X-linked severe combined immunodeficient (SCID-X1) patients (Genovese et al, 2014) has recently been proposed for clinical application.

In none of the gene-editing studies reported to date is a selection advantage expected in the self-renewing HSCs (Genovese et al, 2014; Hoban et al, 2015; De Ravin et al, 2016). In the current study, we therefore propose a gene-editing approach for the treatment of the bone marrow failure (BMF) that takes place in virtually all

1 Division of Hematopoietic Innovative Therapies, Centro de Investigaciones Energéticas, Medioambientales y Tecnológicas, Madrid, Spain
2 Advanced Therapies Unit, Instituto de Investigación Sanitaria Fundación Jiménez Díaz, Madrid, Spain
3 Centro de Investigación Biomédica en Red de Enfermedades Raras, Spain
4 San Raffaele Telethon Institute for Gene Therapy (SR-Tiget), IRCCS San Raffaele Scientific Institute, Milan, Italy
5 Vita Salute San Raffaele University, Milan, Italy
6 Molecular Cytogenetics Group, Human Cancer Genetics Program, Centro Nacional de Investigaciones Oncológicas (CNIO), Madrid, Spain
7 Servicio Hemato-Oncología Pediátrica, Hospital Infantil Universitario Niño Jesús, Madrid, Spain
8 Fundación Investigación Biomédica, Hospital Infantil Universitario Niño Jesús, Madrid, Spain
9 Servicio de Oncología y Hematología Pediátrica, Hospital Universitario Vall d'Hebron, Barcelona, Spain
10 Sangamo Therapeutics, Inc., Richmond, CA, USA
*Corresponding author. Tel: +34913466518; Fax: +34913466484; E-mail: juan.bueren@ciemat.es
**Corresponding author. Tel: +34913460890; Fax: +34913466484; E-mail: paula.rio@ciemat.es

Fanconi anemia (FA) patients (Butturini *et al*, 1994). While the genetic manipulation of HSPCs from FA patients needs to overcome the increased apoptotic predisposition and high sensitivity of these cells to *in vitro* culture, the selective advantage observed in reverted HSPCs from FA mosaic patients (Waisfisz *et al*, 1999; Gregory *et al*, 2001; Gross *et al*, 2002) and also in induced pluripotent stem cells (iPSCs) corrected by lentiviral vectors (Raya *et al*, 2009) suggests that the infusion of a very limited number of gene-edited FA HSCs might be sufficient to progressively repopulate the whole hematopoiesis of these patients.

Since FA can be caused by many different mutations in any of the 22 *FANC* genes discovered so far (Bogliolo & Surralles, 2015; Bluteau *et al*, 2016; Park *et al*, 2016; Knies *et al*, 2017), we decided to develop a versatile gene therapy platform based on the specific integration of a therapeutic transgene into the adeno-associated virus integration site 1 (*AAVS1*) "safe harbor" locus (Lombardo *et al*, 2011) using zinc finger nucleases (ZFNs). Although we initially focused our aims in the specific *AAVS1* targeting of the *FANCA* gene, whose mutations account for about 60% of the FA patients (Casado *et al*, 2007), this approach could be easily applied to the delivery of all the other *FANC* genes.

In a previous study, we used the same strategy for the integration of *FANCA* into patient-derived fibroblasts that were subsequently reprogrammed and differentiated *in vitro* to generate disease-free FA hematopoietic progenitors (Rio *et al*, 2014). Due to current limitations for the use of iPSC-derived HSCs in human hematopoietic therapies (See review in Daniel *et al*, 2016), we have now focused our efforts on the gene editing of primary HSPCs directly obtained from FA patients. Our results demonstrate for the first time the feasibility of conducting therapeutic gene editing in primary CD34+ cells from FA-A patients.

## Results

### Efficient gene targeting of reporter genes in lymphoblastic cell lines from healthy donors and FA patients

Previously we showed the feasibility of targeting a therapeutic *FANCA* expression cassette in the *AAVS1* site of FA-A fibroblasts (Rio *et al*, 2014). Due to the fact that in that study a transient expression of *FANCA* from the non-integrated donor might have facilitated the HDR-mediated insertion of *FANCA* in FA-A cells—with reported HDR defects (Nakanishi *et al*, 2005)—in the current study we first investigated the possibility of targeting a non-therapeutic donor (PGK-EGFP) in *FANCA*-deficient cells, as compared to *FANCA*-proficient cells. Data in Fig 1A show that the ZFN-mediated targeting of the PGK-EGFP donor was feasible in FA-A LCLs. Although the overall proportion of EGFP+ cells obtained after gene editing of FA-A LCLs (mean value: $1.61 \pm 1.51\%$) was apparently lower, either with respect to their complemented FA-A LCLs counterparts ($2.53 \pm 1.97\%$; $P = 0.02$) or HD LCLs (CP1, CP2, C3, and C4; $4.07 \pm 3.89\%$; $P = 0.34$), no significant differences could be observed among the different experimental groups.

In–out PCR analysis of the 5′ and the 3′ ends of the target *AAVS1* site confirmed in all instances the integration of the transgene in the *AAVS1* locus (Fig 1B), supporting the fact that the proportion of EGFP+ cells actually corresponds to the efficiency of gene editing of

these cells. Moreover, the sequencing of PCR products evidenced the proper vector-to-genome junctions not only in HD LCLs, but also in FA LCLs, demonstrating the precise HDR-mediated integration of the donor construct in the *AAVS1* locus, both in HD and in FA LCLs (Fig 1C).

Taken together the results obtained in this first set of experiments demonstrate that FANCA is not essential for the precise HDR-mediated insertion of a transgene in the genome of human FA-A cells, although we cannot discard the hypothesis that it may enhance the efficiency of this process.

### Restored FA pathway in FA-A lymphoblastic cell lines after *FANCA* insertion in the *AAVS1* site

To investigate whether the insertion of the *FANCA* expression cassette in the *AAVS1* locus of FA-A LCLs conferred a phenotypic correction in these cells, two therapeutic IDLV donors were used (Fig 2A). In both vectors, the therapeutic *FANCA* gene was placed under the transcriptional control of the human PGK promoter. In the PGK-*FANCA*/PuroR construct, a downstream 2A PuroR gene was used, while in the EGFP/PGK-*FANCA* construct, a promoterless EGFP cDNA was preceded by a splice acceptor (SA) site and a self-cleaving 2A peptide, to facilitate the identification of gene-edited cells (Rio *et al*, 2014).

When FA-A LCLs were treated both with the ZFNs and the PGK-*FANCA*/PuroR IDLV, a significant decrease in the hypersensitivity of these FA cells to MMC was observed, showing the complementation by gene editing of these cells (Fig 2B). Consistent with this observation, these ZFN-mediated edited cells were also more resistant to puromycin as compared to non-edited cells (survivals to 0.5 μg/ml puromycin in edited and non-edited FA-56 LCLs cells were $7.52 \pm 0.32\%$ and $14.80 \pm 1.16\%$, respectively). Although the ZFN-mediated editing of FA-A LCLs with the EGFP/PGK-*FANCA* donor did not show the expected expression of EGFP (data not shown), these edited cells also showed the correction of their hypersensitivity to MMC (Fig 2B).

As the generation of oxidative reactive species (ROS) is increased in FA cells (Joenje *et al*, 1981), we tested differences in ROS levels both in edited and not edited FA LCLs. As shown in Fig EV1, a significant reduction of ROS was observed when FA-55 LCLs were treated both with the ZFN and the PGK-*FANCA*/PuroR LV, as compared with the parental cells, or cells only transduced with the therapeutic donor.

These results demonstrate that the targeting of therapeutic *FANCA* donors in the *AAVS1* site of FA-A LCLs mediates the correction of two characteristic phenotypes of FA cells, their hypersensitivity to DNA cross-linking drugs and the spontaneous production of ROS.

Since FANCA is essential for FANCD2 foci formation in DNA damaged sites, the percentage of cells with FANCD2 foci was also analyzed in HD LCLs, as well as in FA LCLs (either untreated or treated with MMC). These cells were analyzed in the absence of any treatment, after treatment with the therapeutic donor IDLV, or after treatment with the therapeutic donor and the ZFNs. In all instances, quantification of FANCD2 foci was determined in MMC-untreated and in MMC-treated samples. While a very low number of cells with FANCD2 foci were observed in FA LCLs, either untreated or only treated with the therapeutic donor, gene-edited FA LCLs efficiently

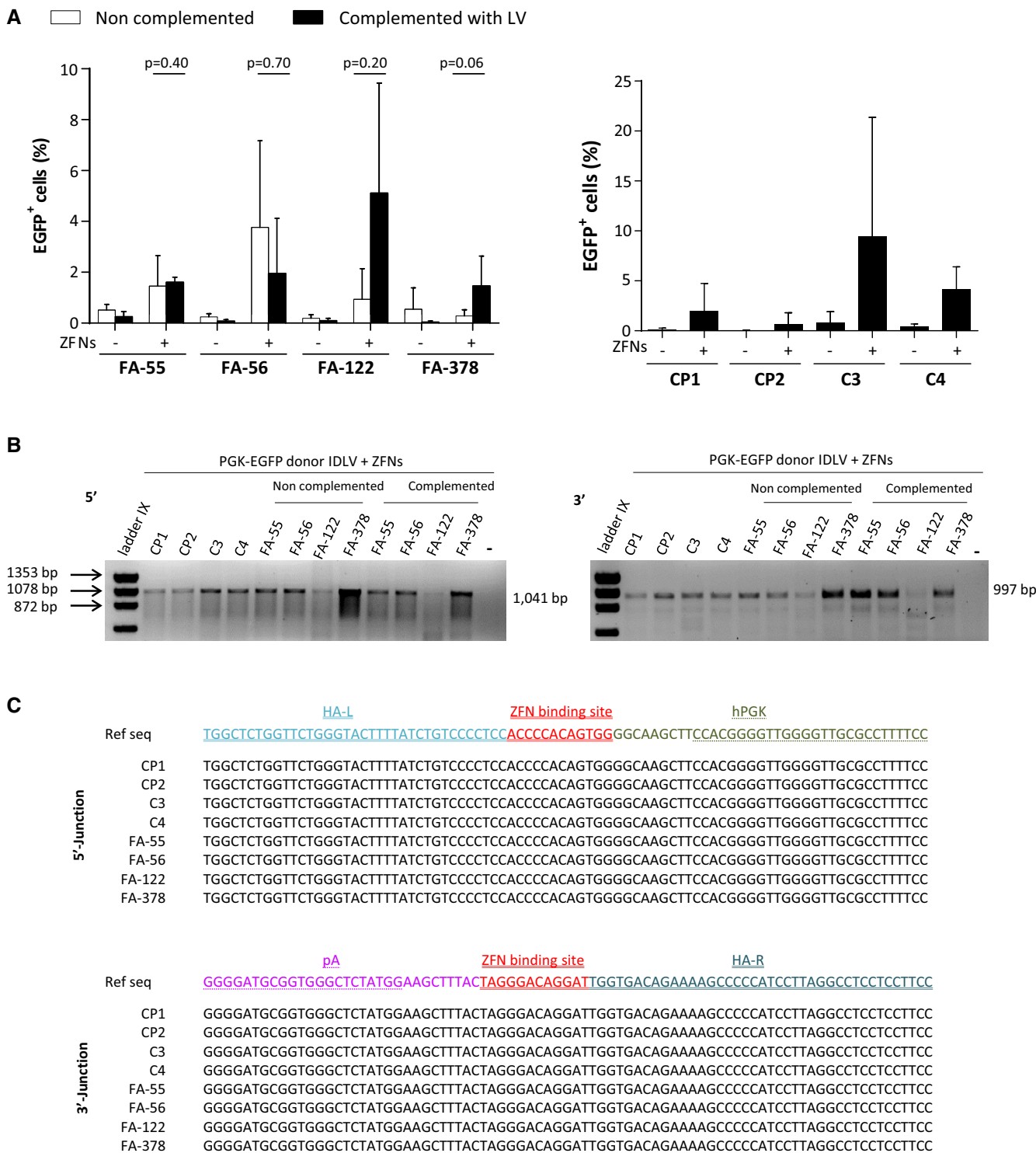

**Figure 1. Efficiency of gene targeting in lymphoblastic cell lines from healthy donors and FA-A patients.**

A  Gene-targeting efficiency in LCLs from FA-A patients before (left panel; white bars) or after complementation with a FANCA lentiviral vector (left panel; black bars), in comparison with LCLs from healthy donors (CP1, CP2, C3 and C4; right panel). Results in the absence (−) or presence of ZFNs (+) are shown. Bars represent mean with SD, $n = 5$. Mann–Whitney $U$-test was used for statistical analysis.

B  In–out PCR analyses of gene-edited LCLs from four different HDs and four FA-A patients. FA LCLs include not complemented and LV-complemented counterparts after selection of gene-edited cells by cell sorting. −: negative control.

C  Sanger sequencing of the 5′ and 3′ junctions amplified by PCR. Reference sequence (Ref seq) is shown. HA-L, left homology arm; HA-R, right homology arm; PGK, human phosphoglycerate kinase promoter; pA, PolyA and ZFN binding site.

Source data are available online for this figure.

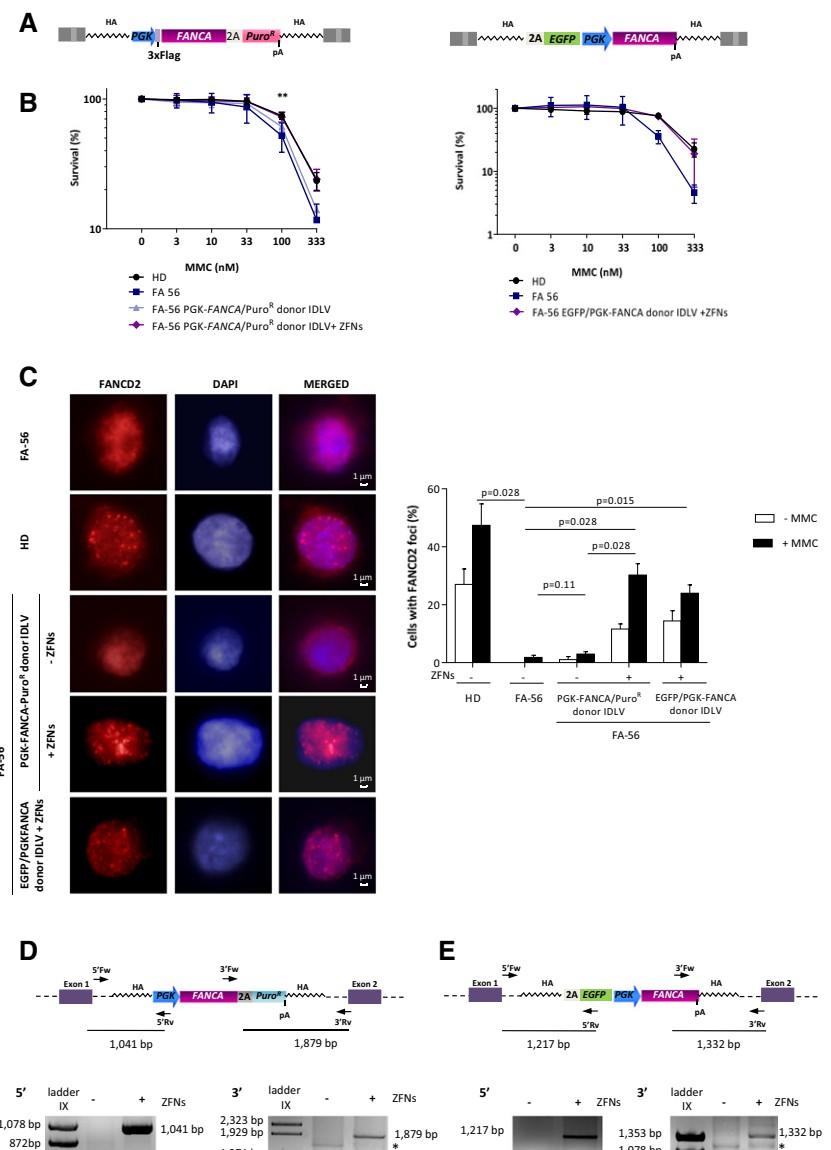

**Figure 2.  Gene targeting with two different therapeutic vectors restores the FA pathway in FA-A LCLs.**

A   Schematic representation of the two therapeutic donor constructs tested. Left: PGK-*FANCA*/Puro[R] donor IDLV. Right: EGFP/PGK-*FANCA* donor IDLV. In both cases, the *FANCA* gene was expressed under the control of the PGK promoter. PGK-*FANCA*/Puro[R] donor IDLV also includes a puromycin resistance gene preceded by the 2A self-cleaving peptide sequence. The EGFP/PGK-*FANCA* donor carries a promoterless EGFP, whose expression is driven by the endogenous promoter of the *PPP1R12C* gene through the self-cleaving peptide sequence, was included as a marker gene. HA, homology arm; pA, polyA.

B   MMC survival curve of FA-56 LCL after gene targeting. Cells were transduced with the PGK-*FANCA*/Puro[R] donor IDLV (left panel), or the EGFP/PGK-*FANCA* donor IDLV (right panel) and then nucleofected with mRNA AAVS1-ZFNs. In left panel, a control transduced with the PGK-*FANCA*/Puro[R] donor IDLV in the absence of ZFNs was also included. HD (C3) and FA-56 LCLs were used as controls. MMC resistance was evaluated 10 days after the drug treatment. Data are shown as mean ± SD. Two-way ANOVA followed by Tukey's *post hoc* test. **$P < 0.01$ (100 nM dose of MMC).

C   Nuclear FANCD2 foci formation in FA-56 LCLs after gene targeting. Left panel: Representative immunofluorescence microphotographs of FANCD2 foci (Texas-Red) counterstained with 4′,6-diamidino-2-phenylindole dihydrochloride (DAPI) in LCLs from: FA-56; HD; FA-56 PGK-*FANCA*/Puro[R] donor IDLV both in the absence (−) or presence (+) of ZFNs and FA-56 EGFP/PGK-*FANCA* donor in the presence (+) of ZFNs. In all instances, images correspond to cells treated with MMC. Right panel: Percentage of cells with FANCD2 foci in the absence or presence of MMC. HD and FA-56 were used as positive and negative controls, respectively, and FANCD2 foci were analyzed. Cells with more than 10 foci were counted as positive cells. Bars represent mean with SD of four different counts. Data are shown as mean ± SD, and *P*-values were calculated using two-tailed paired *t*-test.

D   Schematic representation of the PGK-*FANCA*/Puro[R] cassette integrated in the *AAVS1* locus with 5′ and 3′ primers used for in–out PCR (upper panel). In–out PCR to detect the integration in the *AAVS1* locus in FA-56 after gene editing both for the 5′ and 3′ junctions. Sizes of the expected bands are indicated (bottom panel). *indicates non-specific amplification.

E   Upper panel shows EGFP/PGK-*FANCA* donor cassette integrated in the *AAVS1* locus with 5′ and 3′ primers used for in–out PCR and bottom panel in–out PCR to detect the integration in the *AAVS1* locus in FA-56 after gene editing both for the 5′ and 3′ junctions. Sizes of the expected bands are indicated. *indicates non-specific amplification.

Source data are available online for this figure.

restored the formation of FANCD2 foci, mimicking the behavior of HD LCLs (Figure 2C).

As observed in Fig 1A, in–out PCR analyses confirmed the integration of the therapeutic donors, either the PGK-*FANCA*/Puro^R (Fig 2D) or the EGFP/PGK-*FANCA* (Fig 2E) donors, in the *AAVS1* locus of FA-A cells.

These results demonstrate that the targeted insertion of *FANCA* in the *AAVS1* locus of FA-A LCLs efficiently corrects characteristic phenotypes of these cells.

### Efficient and safe targeting in the *AAVS1* locus of healthy donor cord blood CD34+ cells

To achieve targeted integration into the *AAVS1* site of human HSPCs, cord blood (CB) CD34+ cells were investigated first. Cells were prestimulated for 48 h and then transduced with the IDLV donor. Twenty-four hours later, samples were electroporated with the *AAVS1*-ZFN mRNAs (doses were optimized according to prior analyses; see Fig EV2). To monitor edited cells both *in vitro* and after transplantation into immunodeficient mice, targeting experiments were first performed with the PGK-EGFP donor. Flow cytometry analyses conducted 3 and 10 days after electroporation showed similar values of EGFP+ cells (around 14%) in CD34+ cells treated both with the donor and the ZFNs. In contrast to this data, almost no EGFP+ cells were observed (< 0.6%) when cells were only treated with the PGK-EGFP donor (Fig 3A; representative dot plots are shown in Fig EV3). Analysis of EGFP expression within different CD34+ subpopulations showed the presence of EGFP+ cells in all cell types, including the primitive CD34+/CD133+/CD90+ precursors. An average of 6.9 ± 4.7% of these cells was EGFP+ after 10 days of culture (Fig 3B). Additionally, clonogenic assays showed that 14.1 ± 7.6% of the colonies were EGFP+. In–out PCR analyses confirmed the integration of the donor cassette in the *AAVS1* locus in 90.5% of these colonies. When CB CD34+ cells were only transduced with the PGK-EGFP IDLV, neither EGFP-fluorescent colonies nor integrations in the *AAVS1* locus were observed in any of the 50 colonies analyzed corresponding to five independent experiments (see representative in–out PCR in Fig 3C). Taken together, these experiments demonstrate the specific targeting of the PGK-EGFP donor construct in primary human CD34+ cells.

Although *AAVS1*-ZFNs used in these experiments have been partially modified to improve their specificity (Lombardo *et al*, 2011) as compared to the ones previously described (Mussolino *et al*, 2014), the target sequence was maintained. Therefore, the same five potential off-targets previously studied by Mussolino *et al* (Mussolino *et al*, 2014), namely *ATRNL1*, *BEGAIN*, *CHRAC1*, *H19*, and *LINC00548* were analyzed by deep sequencing. As shown in Table 1, low off-target activity was detected in three of these target loci: *LINC00548, H19* and *BEGAIN* (0.52, 0.17 and 0.02%, respectively), whereas no off-target activity was observed in *ATRNL1* and *CHRAC1* (the detection limit of these analyses was 0.01%). The frequencies of detected INDELs were extremely low as compared to the intended *AAVS1* target site, which showed an INDEL rate of 57.4% (Fig EV2 and Table 1). Moreover, both the *BEGAIN* and *LINC00548* off-target sites occurred in the middle of an intron, whereas the *H19* occurred in an intergenic region. All these observations suggest that no deleterious effects are expected as a

consequence of the off-target activity of these *AAVS1*-ZFNs in human CD34+ cells.

### Targeting of reporter and therapeutic genes in the *AAVS1* locus of healthy donor hematopoietic stem cells

To study the repopulating ability of CD34+ cells edited with the PGK-EGFP donor, 3–5 × 10^5 CD34+ cells treated as in experiments of Fig 3 were transplanted into NSG (black squares) and also into NSG-SGM3 mice (black circles), that facilitate a preferential reconstitution with myeloid cells (Billerbeck *et al*, 2011; Miller *et al*, 2013), which are markedly affected in most FA patients. In these experiments, the efficiency of gene editing determined after 10 days of liquid culture was 11.13 ± 4.87% (not shown). As observed in experiments corresponding to Fig 3C, all EGFP+ colonies generated in methylcellulose showed the expected integration in the *AAVS1* site (Fig 4A).

After transplant, the engraftment of human cells was longitudinally monitored by measuring the percentage of hCD45+ cells in serial bone marrow biopsies. Similarly, the percentage of gene editing was determined by analyzing the proportion of hCD45+ cells that were positive for EGFP expression (Fig 4B). The level of human hematopoietic engraftment observed in NSG-SGM3 mice was higher at the earliest time of analysis, 30 days post-transplantation (median value 37.0% of hCD45+ cells), while maximum levels of engraftment in conventional NSG recipients were observed at the last time point of analysis (median value of 69.8% of hCD45+ cells at 90 days post-transplantation; Fig 4C). Analyses of EGFP expression in engrafted human CD45+ cells showed evident, although variable, levels of EGFP expression along the post-transplantation period (Fig 4D). Overall, a similar proportion of EGFP+ cells was observed in NSG-SGM3 and NSG mice at 90 days post-transplantation (median values of 11.9% and 9.8%, respectively, *P* = 0.66). Interestingly, these results were similar to values obtained in cells that were maintained for only 10 days in culture, suggesting a similar gene-editing efficiency in hematopoietic progenitor cells with respect to more primitive hematopoietic precursors with *in vivo* repopulation capacity.

Consistent with previous studies (Billerbeck *et al*, 2011; Miller *et al*, 2013), NSG-SGM3 mice facilitated the generation of human myeloid cells at the expense of reducing B- and T-cell reconstitution, while human B cells were predominant in NSG mice (Fig 4E). Regardless of the proportion of CD3+, CD19+, CD33+, and CD34+ cells found in recipients' BM and spleen, a significant number of EGFP+ cells were found in all lineages, including CD34+ cells (see Figs 4F and EV4).

In–out PCRs conducted with total BM cells from six mice transplanted with EGFP-edited CD34+ cells showed the expected integration of the *EGFP* cassette in the *AAVS1* site (Fig 4G). To further confirm the targeted insertion in human CFCs generated in the hematopoietic tissues of transplanted mice, BM cells from these animals were seeded in methylcellulose cultures that specifically allowed the growth of human colonies. Once again, in–out PCR analyses conducted in human BM-derived EGFP+ colonies confirmed the integration of the donor construct in the *AAVS1* locus of engrafted human progenitor cells (Fig 4H).

To confirm that gene editing was targeting long term hematopoietic stem cells (LT-HSCs), BM cells from primary recipients, either

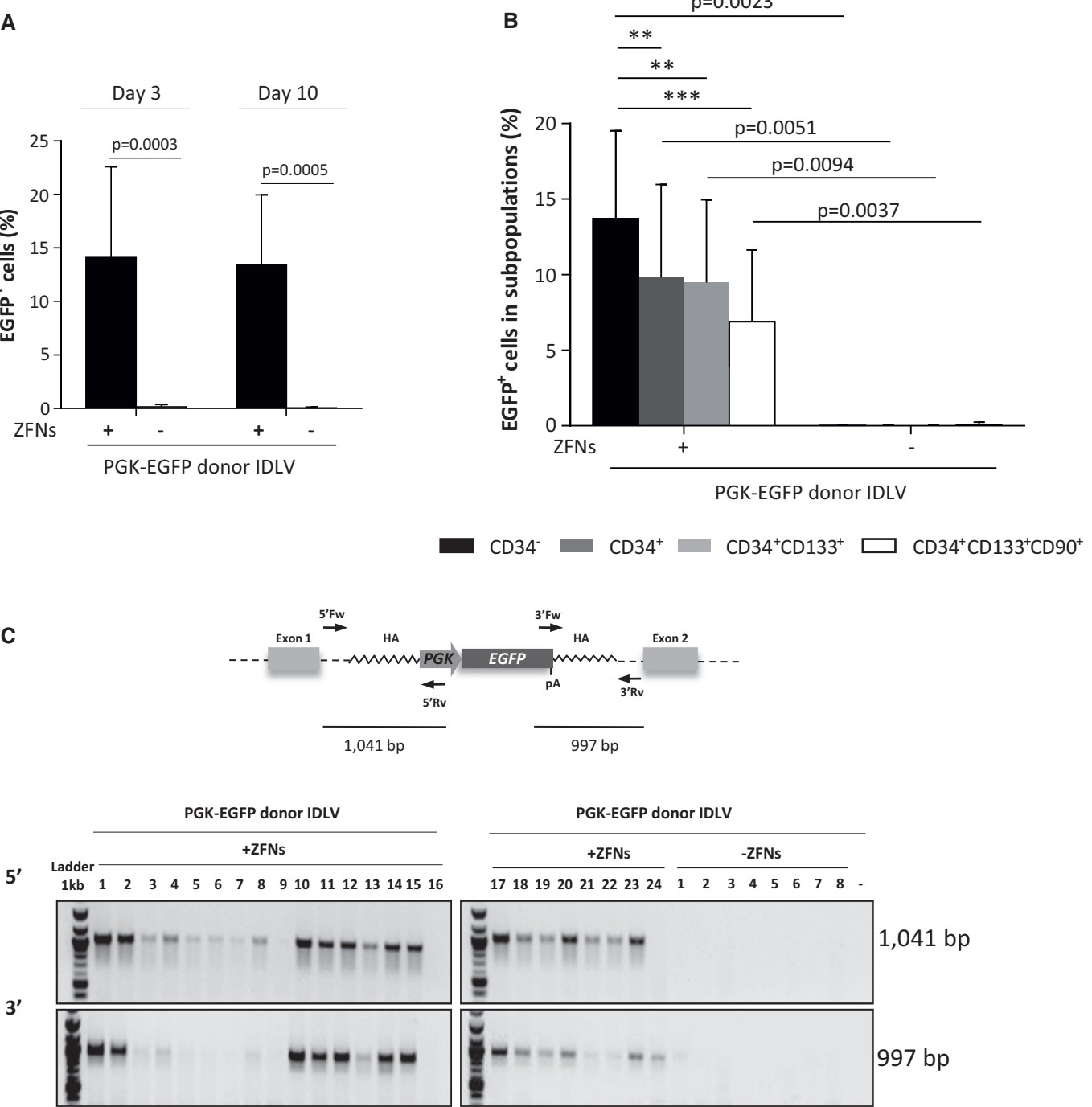

**Figure 3. Gene targeting in healthy donor human CD34+ cells.**

A Gene-targeting efficiency in HD CD34+ cells measured by flow cytometry 3 and 10 days after gene editing. The percentage of EGFP+ cells is shown. Bars represent mean ± SD, $n = 8$. $P$-values were calculated by two-tailed paired $t$-test.

B Gene-targeting efficiency in the different subpopulations of hematopoietic progenitors (differentiated: CD34−, committed progenitors: CD34+, early progenitors: CD34+CD133+ and primitive cells: CD34+CD133+CD90+). The percentage of EGFP+ cells was measured by flow cytometry at day 10 after gene editing. Bars represent mean ± SD, $n = 8$; ANOVA with Bonferroni's multiple comparison test between subpopulations inside the same group (+ZFN $P = 0.0002$; **$P \leq 0.01$; ***$P \leq 0.001$) and two-tailed paired $t$-test between different groups (+ZFNs versus −ZFNs) were used.

C Integration analysis in hematopoietic colonies obtained from CD34+ cells treated with the EGFP donor IDLV together with ZFNs (EGFP+ colonies), or with the EGFP donor IDLV alone. Upper panel: Schematic representation of the PGK-EGFP cassette integrated in the *AAVS1* locus with 5′ and 3′ primers used for in–out PCR. Bottom panel: PCR for the 5′ and 3′ junction. Sizes of the expected bands are indicated. −: negative control.

Source data are available online for this figure.

**Table 1.  INDELs quantification on the target and putative off-target sites of AAVS1 ZFNs in edited CD34⁺ cells.**

| Gene | Chromosome | In/Ex | Homology (%) | ZFN-Dimer | NHEJ (%) |
|------|-----------|-------|--------------|-----------|----------|
| AAVS1 | 19 | Ex | 100 | R-6-L | 57.4 |
| LINC00548 | 13 | In | 91.7 | L-6-R | 0.52 |
| H19 | 11 | Outside | 83.3 | L-5-R | 0.17 |
| BEGAIN | 14 | In | 83.3 | R-6-L | 0.02 |
| ATRNL1 | 10 | Outside | 79.2 | L-5-R | ns |
| CHRAC1 | 8 | Outside | 75 | R-6-L | ns |

Possible ZFN off-target site is located in: (Ex) exon, (In) intron or (Outside) intergenic, closed to the RefSeg gene indicated in the first column. Homology: percentage of sequence identity to the AAVS1-ZFNs binding sites. ZFN-Dimer: All possible off-targets are bound by heterodimeric (L-R and R-L) ZFN pair. Number indicates the spacer length in bp between the ZFN binding sites. NHEJ (%): INDELs quantification expressed as percentage of the total number of reads for each off-target. ns: not significant INDELs detected (according to the two criteria stablished in the low-frequency variant calling (see Materials and Methods for further information). The % of INDELs in AAVS1 locus has been estimated by Surveyor assay (Fig EV1).

NSG or NSG-SGM3 mice, were re-transplanted into secondary NSG recipients. Bone marrow cells from primary NSG but not from NSG-SGM3 mice repopulated secondary recipients, suggesting a stressed differentiation of human HSCs in NSG-SGM3 mice (Fig EV5A). The analysis of human hematopoiesis in secondary recipients showed the presence of myeloid, B lymphocytes, and T lymphocytes. Significantly, the presence of EGFP⁺ cells was also detected in human engrafted cells, although at lower levels, around 1% (Fig EV5B–D), with respect to values observed in primary recipients.

Taken together, the *in vivo* experiments conducted with HD CB CD34⁺ cells that were edited with a non-therapeutic donor demonstrated the targeting of primitive human HSCs characterized by long-term multipotent *in vivo* repopulating properties.

In the next set of experiments, we moved on to investigate the efficacy of gene editing, now using a therapeutic donor: the EGFP/PGK-*FANCA* donor. Consistent with data obtained in FA LCLs, no EGFP-positive cells were detected as early as 10 days after the *in vitro* culture of cells edited with this donor, in which EGFP expression is driven from the regulatory signals present in the *AAVS1* locus. This contrasts with the significant number of CD34⁺ cells that were positive for EGFP expression after gene editing with the PGK-EGFP donor (Fig EV6A). Because in–out PCR analyses conducted in hematopoietic colonies evidenced the integration of the EGFP/PGK-*FANCA* donor in the *AAVS1* locus (Fig EV6B), these results suggest that the endogenous regulatory sequences of the *AAVS1* locus are not sufficiently active to promote detectable EGFP expression in human hematopoietic cells.

To investigate the repopulating ability of HD CD34⁺ cells edited with this therapeutic cassette, samples were transplanted into NSG-SGM3 mice. As observed after gene editing with the EGFP donor, analysis of the hematopoietic organs of transplanted mice confirmed a multilineage human hematopoietic engraftment of these animals (Fig EV6C–E). qPCR analyses confirmed the integration of the EGFP/PGK-*FANCA* cassette both in the BM and in the spleen of these mice (data not shown), demonstrating the feasibility of targeting a therapeutic FA donor in the *AAVS1* site of HD HSPCs characterized by *in vivo* repopulating properties.

### Targeted mediated gene therapy in CD34⁺ cells from FA-A patients

Having demonstrated the specific insertion of marker and FA therapeutic cassettes in the *AAVS1* locus of primary HD HSPCs, we investigated the feasibility of correcting the phenotype of CD34⁺ cells from FA-A patients using the same gene-editing approach. To achieve this aim, small aliquots of mobilized peripheral blood (mPB) CD34⁺ cells from five FA-A patients were treated with the therapeutic donors and the *AAVS1* ZFNs, as was done in experiments with HD CB CD34⁺ cells.

To test the phenotypic correction of FA CD34⁺ cells after gene targeting, cells were plated in methylcellulose both in the absence and in the presence of MMC, and colonies were scored after 14 days of culture. Mean survivals to MMC corresponding to edited CD34⁺ cells were $29.91 \pm 24.63\%$, which were significantly increased with respect to the MMC resistance observed in cells only transduced with the therapeutic donor (mean value: $2.98 \pm 4.46\%$; $P = 0.016$; Fig 5A). These results suggest the phenotypic correction of FA-A primary CD34⁺ cells by gene editing. To confirm that the increased number of MMC-resistant colonies was due to the targeted integration of the *FANCA* cassette in the *AAVS1* locus, in–out PCRs from a pool of hematopoietic colonies grown in the absence and presence of MMC were performed in samples from two FA-A patients. In the case of patient FA-655, a specific integration of the therapeutic cassette was observed in MMC-selected samples, but not in unselected cells. Notably, in patient FA-712 a faint band could be detected even in the absence of MMC-selected cells, although a marked increase in the intensity of this band was detected after MMC selection. The analysis of individual colonies picked from patient FA-655 showed that 10% of colonies grown in the absence of MMC were positive for the integration of *FANCA* in the *AAVS1* site (See representative in–out PCR in Fig 5C). This final experiment demonstrates for the first time the feasibility of correcting the disease phenotype of primary CD34⁺ cells from FA-A patients by targeted gene therapy.

## Discussion

The development of engineered nucleases and donor constructs capable of inserting DNA sequences in specific sites of the cell genome is opening up new possibilities for therapeutic gene editing of inherited and acquired diseases (reviewed in Corrigan-Curay *et al*, 2015; Hoban & Bauer, 2016; Osborn *et al*, 2016). The efficacy and specificity of ZFNs designed for the targeting of the *AAVS1* "safe harbor" locus, together with the high number of private mutations characterized in FA-A patients, suggest that the specific insertion of *FANCA* in the *AAVS1* site of FA-A patient's

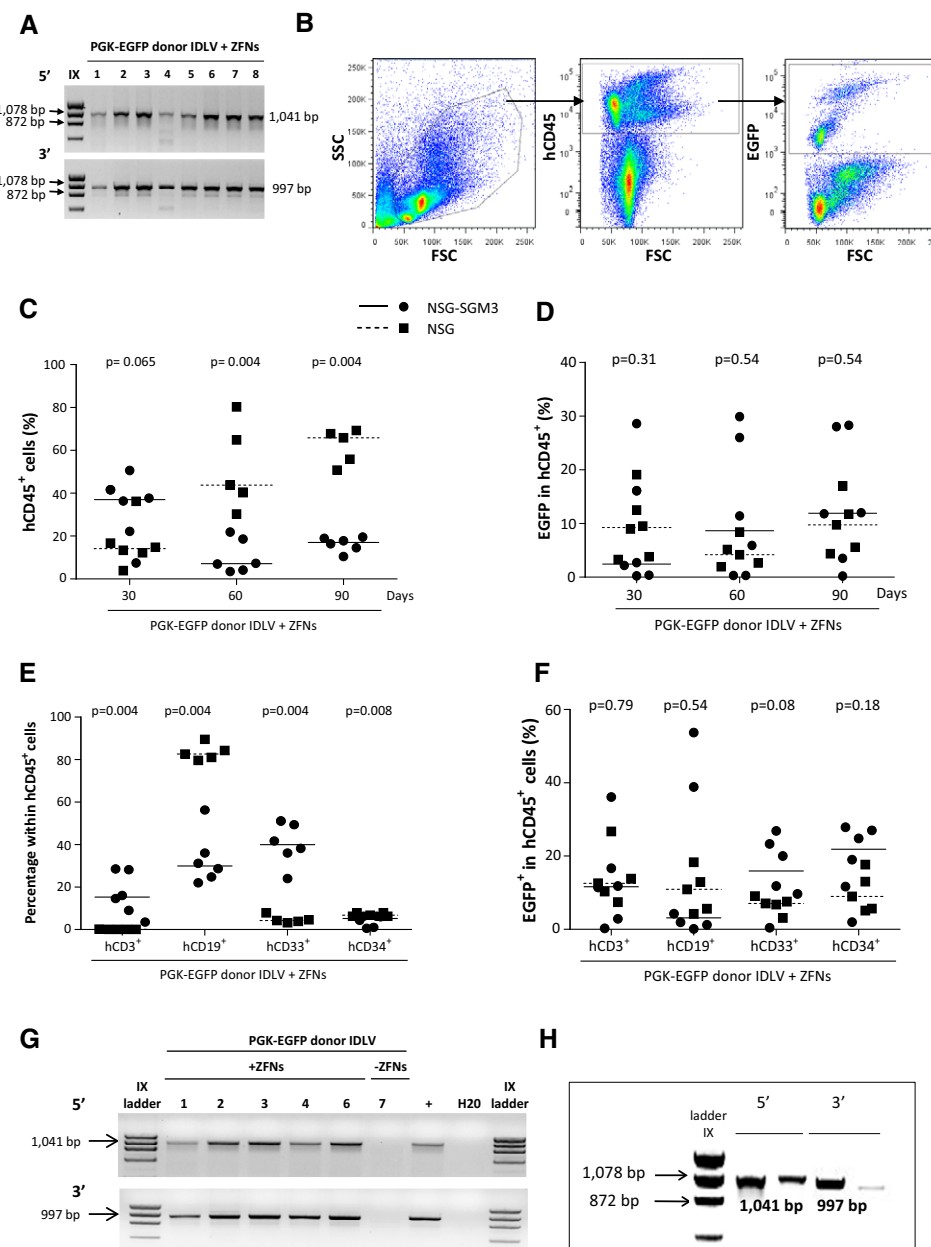

**Figure 4.  Repopulating capacity of gene-edited CD34+ cells.**

A   In–out PCR analysis for the 5′ and the 3′ ends of the target *AAVS1* site in EGFP+ hematopoietic colonies derived from gene-edited CD34+ cells used in immunodeficient mice transplantation.

B   Representative dot plot analysis of the bone marrow from an immunodeficient mouse transplanted with gene-targeted cells. From left to right: total BM, hCD45+ cells and finally EGFP+ cells in the hCD45+ population are shown.

C   Human engraftment measured as the % of hCD45+ cells. Squares: individual mice transplanted with CD34+ cells in NSG mice. Circles: individual mice transplanted with CD34+ cells in NSG-SGM3 mice. Dashed line indicates median value in NSG mice and solid line median value in NSG-SGM3 mice.

D   Percentage of EGFP+ cells in human CD45 population at 30, 60, and 90 days post-transplantation.

E   Multilineage BM reconstitution was evaluated by flow cytometry using CD3 antibody for T cells, CD19 for B cells, CD33 for myeloid cells, and CD34 for hematopoietic stem and progenitor cells 90 days post-transplantation.

F   Percentage of EGFP+ cells in the different subpopulations of the BM.

G   In–out PCR to detect the integration of the HDR cassette in the AAVS1 locus in BM obtained from mice 90 days post-transplantation (lanes 1–6: mice transplanted with PGK-EGFP IDLV donor and AAVS1-ZFNs mRNA; lane 7: mouse transplanted with CD34+ cells only transduced with the PGK-EGFP IDLV donor; +: LCL edited with the same approach). PCR analysis showed a 1,041-bp band corresponding to the 5′ junction and a 997-bp band corresponding to the 3′ junction.

H   Analysis of the integration in the *AAVS1* locus in two different EGFP+ CFCs obtained from primary mice transplanted with gene-edited cells.

Data information: Data are represented as median values (dashed lines indicate median value in NSG mice and solid lines median value in NSG-SGM3 mice). Statistical analysis was conducted using Mann–Whitney test (C–F). *n* = 5 NSG mice and *n* = 6 NSG-SGM3 mice.

Source data are available online for this figure.

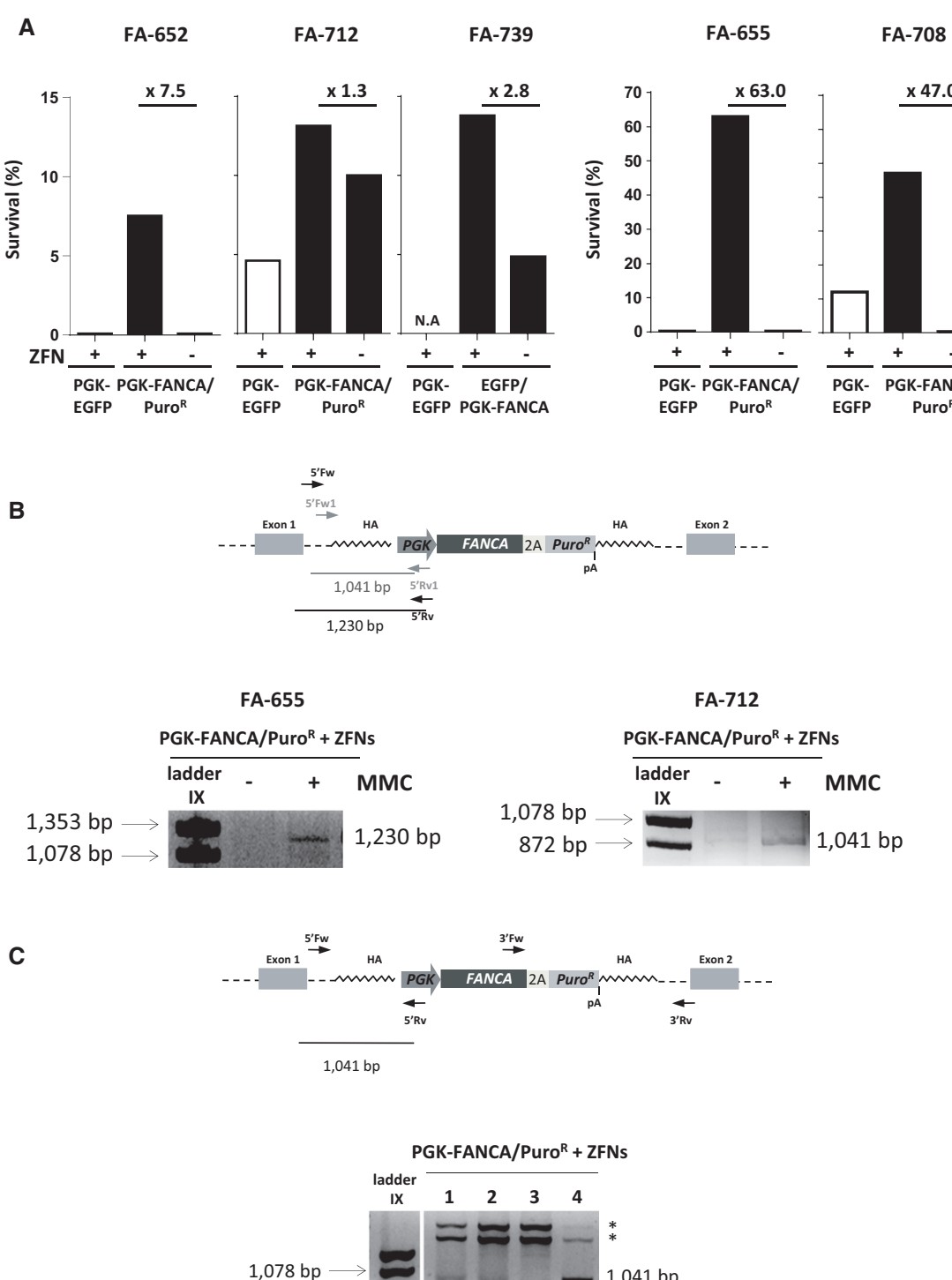

**Figure 5.  Restoration of resistance to MMC in HSPCs from five different FA-A patients after gene editing.**

A    MMC survival in HSPCs from FA-A patients plated using two different therapeutic donor IDLVs (EGFP/PGK-*FANCA* donor and PGK-*FANCA*/Puro^R donor) combined with ZFNs or not (black bars). A PGK-EGFP donor IDLV together with ZFNs was used as a control (white bars).

B    Upper panel shows the schematic representation of the PGK-*FANCA*/Puro^R cassette integrated in the *AAVS1* locus with the different primers used for in–out PCR. Bottom panels: Integration analysis by PCR in a pool of hematopoietic colonies obtained from patient FA-655 (left) and patient FA-712 (right) after gene targeting using PGK-*FANCA*/Puro^R donor in the absence or presence of MMC. Sizes of the expected bands are indicated.

C    Schematic representation of the PGK-*FANCA*/Puro^R cassette integrated in the *AAVS1* locus (upper panel) showing primers used for in–out PCR conducted in individual colonies from patient FA-655 obtained after gene editing using the PGK-*FANCA*/Puro^R donor IDLV together with *AAVS1*-ZFNs (bottom panel). *indicates non-specific band. Size of the expected band is indicated.

Source data are available online for this figure.

HSPCs would result in a particularly useful platform for restoring normal hematopoiesis in these patients. Although recent results have shown the efficiency of AAVs to deliver donor cassettes in HSPCs (Wang *et al*, 2015), the large packaging capacity of IDLVs and the high transduction efficacy of these vectors on human HSPCs (Lombardo *et al*, 2007) supported the use of IDLVs in our gene-editing studies to facilitate the delivery of the therapeutic *FANCA* gene together with additional marker genes in these cells.

Previous studies have demonstrated that gene editing is feasible in CD34+ cells from patients with hematological disorders such as SCID-X1 (Genovese *et al*, 2014) or chronic granulomatous disease (CGD) (De Ravin *et al*, 2016). Nevertheless, the editing of CD34+ cells from FA patients, where the number and function of these cells is highly compromised (Ceccaldi *et al*, 2012), has not been attempted so far. Additionally, gene editing in FA cells is challenging owing to previous observations showing the involvement of the FA pathway in HDR (Nakanishi *et al*, 2005). Despite these limitations, the proliferation advantage expected as a result of the therapeutic editing of FA HSPCs may constitute a characteristic advantage of FA compared to many other inherited disorders in which the HSPCs do not reveal any phenotypic defect.

Although our previous study demonstrated the possibility of conducting gene editing in FA-A fibroblasts, this could be a consequence of the potential transient expression of FANCA driven by the therapeutic donor prior to integration in the cell genome (Rio *et al*, 2014). Our results in this new study confirm in LCLs from four different FA-A patients the feasibility of conducting gene editing, even with a non-therapeutic donor that only carries a reporter EGFP transgene, demonstrating for the first time that *FANCA* is not essential for the HDR-mediated editing of the human genome. Nevertheless, our data also showed that the efficacy of gene editing in not-complemented FA-A cells was two- to threefold lower as compared to either their complemented cell counterparts or to HD LCLs. Although these differences did not reach statistical significance, our results are consistent with previous data obtained in different experimental models showing that, in contrast to other FA proteins such as *BRCA2*, *FANCA* is only partially involved in HDR (Nakanishi *et al*, 2005, 2011).

Significantly, when donors carrying the therapeutic *FANCA* gene (PGK-*FANCA*/Puro^R or EGFP/PGK-*FANCA*-IDLVs) were used for the editing of FA-A LCLs, an efficient phenotypic correction of these cells was deduced from their decreased MMC-hypersensitivity and production of ROS, and also their restored ability to translocate FANCD2 to DNA damage repair foci. In spite of the phenotypic correction of FA cells edited with the EGFP/PGK-*FANCA* donor, these cells did not show the expected EGFP-expression, suggesting the low activity of the *AAVS1* regulatory sequences in FA LCLs, which contrasts with our results in FA-A fibroblasts edited with the same construct (Rio *et al*, 2014). This observation reinforced the relevance of including an exogenous promoter in therapeutic donors to be inserted in this locus, at least in FA hematopoietic cells. In the other therapeutic donor (the PGK-*FANCA*/Puro^R-IDLV), the Puro^R gene was included in the cassette to facilitate the selection of edited cells. Although puromycin resistance could be confirmed in edited FA cells, because of the inherent hypersensitivity of parental FA cells to MMC, edited FA cells were very efficiently selected with this drug and thus was routinely used in our experiments.

In our studies conducted with primary HD CD34+ cells, gene-editing efficacies around 14% were achieved using *AAVS1* mRNA

ZFNs and EGFP IDLV donors. Moreover, off-target analyses confirmed the high specificity of the *AAVS1*-ZFNs in these cells, as demonstrated by the 100-fold increase in the targeting of the *AAVS1* site, compared to the off-target sites. Analyses of mice transplanted with gene-edited CD34+ cells also demonstrated the *in vivo* repopulating capacity of these samples. In this respect, while rapid engraftments were noted in NSG-SGM3 mice, higher long-term engraftment levels were generally achieved in NSG recipients. Remarkably, regardless of the strain of immunodeficient mouse used in our experiments, most of the primary transplanted recipients showed the presence of human hematopoietic cells that expressed significant levels of the EGFP transgene, both in myeloid and T and B lymphoid lineages, and even in CD34+ cells. Additionally, the expected insertion of the transgene in the *AAVS1* locus of these cells was demonstrated in our experiments. Interestingly, secondary transplants in NSG mice demonstrated the long-term repopulating capacity of gene-edited cells. Nevertheless, consistent with previous studies (Genovese *et al*, 2014), a decreased efficacy of gene editing was observed in the very primitive HSCs capable of repopulating secondary recipients.

In our final set of experiments, we demonstrated for the first time the gene editing of mobilized PB CD34+ cells directly obtained from FA-A patients, the target population currently considered for the gene therapy of several inherited diseases, including FA (see review in Tolar *et al*, 2012). Moreover, our data showed that the integration of the therapeutic transgene in the *AAVS1* site of FA-A CD34+ cells resulted in the correction of the characteristic MMC-hypersensitivity of these cells, indicating the possibility of conducting therapeutic gene editing in CD34+ cells from FA-A patients.

Although efficacies of gene targeting achieved in FA CD34+ cells are still significantly lower compared to the transduction efficacy achieved by lentiviral vectors (Jacome *et al*, 2009; Becker *et al*, 2010; Gonzalez-Murillo *et al*, 2010), our results suggest that gene editing could be applicable to BMF syndromes such as FA, in which a selection advantage of edited cells could take place, as we already demonstrated in primary fibroblasts and iPSCs from FA patients (Rio *et al*, 2014).

The possibility of harvesting significant numbers of HSPCs from FA-A patients with improved mobilizing drugs (See review in Adair *et al*, 2016) together with the development of efficient gene-editing tools (Tebas *et al*, 2014) suggests that gene editing may constitute a realistic approach for the treatment of FA in the future. Indeed, the use of drugs such as aryl hydrocarbon receptor antagonist (StemRegenin, SR1) or 16,16-dimethyl-prostaglandin E2 (dmPGE2) capable of improving the engraftment of gene-edited HSCs in transplanted recipients (Genovese *et al*, 2014), or molecules capable of enhancing the efficacy of HDR (Zhang *et al*, 2016) may further aid in the development of therapeutic gene editing for the treatment of inherited diseases affecting the hematopoietic system, including FA.

# Materials and Methods

### Cell lines

293T (ATCC: CRL-11268) were used for the production and titration of the integrase-defective lentiviral vectors (IDLVs). Cells were grown in Iscove's modified Dulbecco's medium GlutaMAX™

(IMDM; Gibco) supplemented with 10% Hyclone (GE Healthcare) and 1% penicillin/streptomycin solution (Gibco). Lymphoblastic cell lines (LCLs) were generated by Epstein–Barr virus transformation of peripheral blood B cells from healthy donors or FA patients. FA-A patients were classified by retroviral cell complementation as previously described (Casado *et al*, 2007). FA-A patients and healthy donors were encoded to protect their confidentiality and informed consent was obtained in all cases. Mutations described in FA-A LCLs (Castella *et al*, 2011) have been included in Table EV1.

### Hematopoietic progenitor cells from healthy donors and FA-A patients

Healthy donor CD34+ cells were provided by the Centro de Transfusiones de la Comunidad de Madrid from umbilical cord blood samples (CB) scheduled for discard, after written informed consent from the mothers. Mononuclear cells from pooled CBs were obtained by fractionation in Ficoll–Hypaque according to manufacturer's instructions (GE Healthcare). Purified CD34+ cells were obtained using the MACS CD34 Micro-Bead kit (Miltenyi Biotec). Cells were grown in StemSpam (StemCell Technologies) supplemented with 1% GlutaMAX™ (Gibco), 1% penicillin/streptomycin solution (Gibco), 100 ng/ml SCF and Flt3-ligand, and 20 ng/ml TPO and IL6 (all from EuroBiosciences).

A small number of mobilized peripheral blood (mPB) CD34+ cells from FA-A patients that remained in cell collection bags and tubes from the CliniMACS® System (Miltenyi Biotec), used for the collection (FANCOSTEM trial; NCT02931071) and the subsequent transduction of CD34+ cells with lentiviral vectors (LVs) (FANCOLEN Trial: NCT03157804), was used after the informed consent was signed and after the approval of the corresponding ethics committees. Cells were grown in StemSpam (StemCell Technologies) supplemented with 1% GlutaMAX™ (Gibco), 1% penicillin/streptomycin solution (Gibco), 100 ng/ml SCF and Flt3-ligand, 20 ng/ml TPO and IL6 (all from EuroBiosciences), 10 μg/ml anti-TNF-α (Enbrel-Etanercept), and 1 mM *N*-acetylcysteine (Pharmazam) under hypoxic conditions (5% of $O_2$).

### Vectors

pCCL.sin.cPPT.*AAVS1*.PGK.*FANCA*.2A.Puromycin.pA donor transfer LV (*FANCA*-Puro$^R$ donor IDLV) was generated using elements from the backbones pCCL.sin.cPPT.*AAVS1*.loxP.SA.2A.GFP.pA. loxP.PGK.*FANCA*.pA.Wpre (Rio *et al*, 2014) and pCCL.sin.cPPT. *AAVS1*.PGK.GFP.pA (Lombardo *et al*, 2011). The integrase-defective third-generation packaging plasmid pMD.Lg/pRRE.D64Vint was used to produce IDLV particles (Lombardo *et al*, 2007). ZFN-L and ZFN-R targeting intron 1 of the *PPP1R12C* gene (Genovese *et al*, 2014) were expressed from *in vitro* synthetized mRNA 5X MegaScript Kit T7 Kit (Ambion), 3′-0-Me-m7G(5′)ppp(5′)G RNA Cap Structure Analog (ARCA) (New England Biolabs), and Poly (A) Tailing Kit (Ambion) according to manufacturer's instructions.

### Gene-targeting experiments

Lymphoblastic cell lines from healthy donors or FA-A patients were transduced with the corresponding IDLV donors [multiplicity of infection (MOI): 50 transducing units/cell (TU/cell)] in 96-well culture plates in a final volume of 100 μl. One day after transduction, cells were collected, washed, and nucleofected with 1.5–6 μg of total *in vitro* synthetized mRNA (0.75–3 μg of each ZFN pair), using the SF Cell Line 4D-Nucleofector X Kit (Lonza).

Purified CD34+ cells from healthy donor CBs or from FA-A patients were prestimulated for 48 or 24 h, respectively, as described above, at a density of $10^6$ cells/ml. Thereafter, $10^5$ cells were transduced with the corresponding IDLV donor at a MOI of 100 TU/cell in a 96-well culture plate in a final volume of 100 μl. One day after transduction, cells were collected, washed, and nucleofected with 6 μg of total *in vitro* synthetized mRNA (3 μg of each ZFN pair) using the P3 Primary Cell 4D-Nucleofector X Kit (Lonza). One day after nucleofection, gene-edited cells were assessed either in clonogenic assays or in transplantation experiments.

In some experiments, cells nucleofected with the PGK-FANCA/Puro$^R$ donor IDLV were selected with 0.5 μg/ml of puromycin for 48 h. Cell viability was analyzed by flow cytometry as previously established for MMC survival assay.

### Flow cytometry and cell sorting

Gene-targeted cells were analyzed by flow cytometry (LSR Fortessa; Becton Dickinson Pharmingen). Immunophenotypic analysis of the hematopoietic differentiated cells was performed using CD34 PE-Cy7 (eBioscience), CD133 PE (Miltenyi Biotech) and CD90 APC (Becton Dickinson Pharmingen) antibodies according to the manufacturer's instructions:

Fluorochrome-matched isotypes were used as controls. 4′,6-diamidino-2-phenylindole (DAPI; Roche)-positive cells were excluded from the analysis. Analysis was performed using FlowJo software v7.6.5.

Cell sorting was also used to select gene-edited LCLs by the selection of EGFP+ cells in a BD Influx cell sorter (BD Biosciences).

### Surveyor nuclease assay

DNA was extracted using NucleoSpin® Tissue kit (Macherey-Nagel). Then, a PCR was performed to amplify the region of the *AAVS1* locus in which the ZFNs cut using the primers: NHEJ Fw (5′ CTTCAGGACAGCATGTTTGC 3′) and NHEJ Rv (5′ ACAG-GAGGTGGGGGTTAGAC 3′). The PCR was performed as follows: 10 min 95°C, 40 cycles of 45 s at 95°C, 45 s at 62°C, and 45 s at 72°C, and finally one step for 10 min at 72°C. The 224-bp PCR product was then dehybridized and rehybridized in order to obtain the heteroduplexes. If an INDEL (integration or deletion) has occurred, the corresponding hairpin can be recognized by the Surveyor® nuclease (Survor® mutation detection kit; IDT), generating a band pattern that was visualized on 10% TBE gels 1.0 mm (Invitrogen) and analyzed using ImageJ software. Percentages of cleavage were determined using the equation:

$$\%\text{NHEJ} = \frac{\text{Cleavaged bands} - (2 \times \text{Background})}{(\text{Cleavaged bands} + \text{Parental band}) - (3 \times \text{Background})} \times 100.$$

### In–out PCR

For PCR analysis, genomic DNA from bulk populations was extracted using the NucleoSpin Tissue kit (Macherey-Nagel). Single colonies were pelleted and resuspended in 10 μl of PBS, and genomic DNA was extracted as previously described (Charrier *et al*, 2011).

To detect the targeted integration of the HDR cassette in the *AAVS1* locus, two different pairs of primers for the 3′ or the 5′ integration junction were used (Table EV2).

### Deep sequencing of putative *AAVS1* off-targets

Among the different *AAVS1* off-targets that were identified by Mussolino *et al* (2014) using the bioinformatic tool PROGNOS, we studied five that have also been experimentally validated: ATRNL1, BEGAIN, CHRAC1, H19, and LINC00548.

Genomic DNA from CD34+ cells targeted with the PGK-EGFP IDLV, together with *AAVS1*-ZFNs, was extracted using the NucleoSpin® Tissue kit, and the putative off-target sites were amplified by PCR, generating amplicons of 270 bp surrounding the potential ZFN binding site (Table EV3).

PCR products were purified using the AxyPrep PCR Clean-Up (Axygen), quantified using a Qubit fluorometer (Thermo Fisher Scientific), and used for library construction using the KAPA Library preparation Kit (Kapa Biosystems) for Ion Torrent platforms. The generated DNA fragments (DNA libraries) were sequenced on the Ion Torrent PGM platform, using 400-bp single-end sequencing reads.

### Analysis of *AAVS1* off-target sites

The analysis of the raw sequence data obtained by Ion Torrent PGM platform sequencing was carried out using CLC Genomics Workbench 9.0.1, and those high-quality sequencing reads were mapped against the reference sequences using a length and a similarity fraction of 0.80.

A low-frequency variant calling was performed, obtaining a list of variants with a low representation within each sample following two criteria: a minimum frequency of 0.01% (and a probability of 0.0001) and a forward/reverse sequence reads balancement ≥ 30% combined with a localization in homopolymeric regions ≤ 2 nucleotides.

Those sequences containing INDELs of ≥ 1 bp located within a region encompassing the spacer were considered as ZFN-induced genome modifications.

### Transplantation of gene-edited cells

Healthy donor cord blood (HD CB) CD34+-treated cells ($3–5 \times 10^5$) were intravenously injected either into NOD.Cg-*Prkdc-scid*Tg (hSCF/hGM-CSF/hIL3; NSG-SGM3) or into conventional NOD-SCID-Il2rg$^{-/-}$ (NSG) mice previously irradiated with 1.5 Gy. In all cases, female mice aging 8–12 weeks old were used. Experiments were performed in accordance with the EU guidelines upon approval of the protocols by the Environment Department in Comunidad de Madrid, Spain (Authorization code PROEX: 070/15).

To evaluate the level of human hematopoietic engraftment, cells were collected by femoral bone marrow aspiration at days 30 and 60 post-transplantation. Mice were sacrificed at 90 days post-transplantation, and cells were collected from their femora, tibiae, and spleen. Secondary transplants were conducted by transplanting 80% of femoral BM cells from each primary recipient into one secondary recipient mouse. In all instances, NSG mice irradiated with 1.5 Gy were used as recipients for secondary transplants. At the final time point of analysis, the presence of cells corresponding to the different hematopoietic lineages was determined. Cells obtained from BM aspirates were stained with hCD45 APC (eBioscience) and CD34 PE (BD Biosciences) antibodies according to the manufacturer's instructions. Analyses of multilineage engraftment were performed using CD45 APC-Cy7 (BioLegend), CD34 APC (BD Biosciences), CD33 PE-Cy7 (BD Biosciences), CD19 PE-Cy7, and CD3 APC (BioLegend), according to the manufacturer's instructions.

Fluorochrome-matched isotypes were used as controls. 4′,6-diamidino-2-phenylindole (DAPI)-positive cells (Roche) were excluded from the analysis. All flow cytometric analyses were performed on the LSR Fortessa (BD Biosciences) and analyzed with FlowJo software v7.6.5.

### Analysis of FA pathway functionality

FANCD2 foci were analyzed by immunofluorescence of LCLs treated for 16 h with 40 nM of MMC. After MMC treatment, cells were counterstained with rabbit polyclonal anti-FANCD2 (Abcam, ab187-50) and DAPI as previously described (Hotta & Ellis, 2008; Raya *et al*, 2009). Cells with more than ten foci per cell were scored as positive.

To test the MMC sensitivity of FA LCLs, $5 \times 10^4$ LCLs were seeded in 24-well plates in a final volume of 1 ml and increasing concentrations of MMC were added (from 0 to 333 nM). Cell viability was determined 10 days afterward by flow cytometry using the DAPI vital marker in a LSR Fortessa. Off-line analyses were performed with the FlowJo software v10.

### Analysis of radical oxygen species (ROS)

The production of ROS was measured by flow cytometry using a non-fluorogenic probe (CellROX®Deep Red Reagent; Life Technologies) that upon oxidation exhibits strong fluorogenic signal (640-nm excitation and 665-nm emission) following manufacturer's instructions. Routinely, $5 \times 10^4$ cells were incubated with CellROX® Deep Red Reagent at a final concentration of 5 μM in PBS for 20 min at 37°C. After incubation, cells were washed with PBA and DAPI was added. Quantification of ROS production was conducted in LSR Fortessa and data were analyzed with FlowJo software v7.6.5.

### Clonogenic assays

Clonogenic assays were established using 900 CD34+ HD CB cells or 10,000 mPB CD34+ cells from FA-A patients in enriched methylcellulose medium (StemMACS™ HSC-CFU complete with Epo, Miltenyi Biotech). To test the MMC sensitivity of hematopoietic progenitors from FA-A patients, 3–10 nM of MMC was added to these cultures.

## The paper explained

### Problem

Thanks to the development of designed nucleases, gene editing has become a good alternative to conventional gene therapy in both inherited and acquired diseases. Nevertheless, the possibility of conducting gene editing in HSPCs is still a major challenge, particularly in diseases such as Fanconi anemia (FA), which are characterized by evident phenotypic defects at the HSPC level. Despite these difficulties, the possibility that FA HSPCs treated by therapeutic gene editing may develop *in vivo* proliferation advantage over uncorrected cells is of particular relevance in the field of gene editing.

### Results

We initially showed specific ZFN-mediated targeting of donor constructs in the *AAVS1* "safe harbor" locus in lymphoblastic cell lines (LCLs) from FA-A patients. In this study, we demonstrate that this process is feasible, not only using a therapeutic donor, but also when non-therapeutic donors are used, indicating that FANCA is not essential for the homologous directed repair (HDR)-mediated gene editing of human cells. The specific insertion of *FANCA* in the *AAVS1* site of FA-A LCLs conferred an evident phenotypic correction of these cells, demonstrating therapeutic gene editing in FA-A hematopoietic cells.

Prior to developing therapeutic gene editing in CD34+ cells from FA patients, an efficient protocol aiming at facilitating the insertion of donor constructs in HSPCs from healthy donors was developed. Efficiencies of HDR around 14% were achieved in *in vitro* cultured cells and around 10% and 1%, respectively, in HSCs capable of repopulating the hematopoiesis of primary and secondary immunodeficient recipients.

In our final set of experiments, we demonstrated that the proposed gene-editing approach was capable of correcting the phenotype of hematopoietic CD34+ cells from FA-A patients, showing for the first time the possibility of conducting therapeutic gene editing in HSPCs from patients with a syndrome characterized by bone marrow failure and DNA repair defects.

### Impact

Our results show for the first time the phenotypic correction of primary HSPCs from FA-A patients by means of the specific insertion of a therapeutic FA gene into a "safe harbor" locus, thus minimizing risks of insertional oncogenesis. Our study constitutes the first proof of concept showing the feasibility of conducting gene editing in primary FA CD34+ cells as a future therapeutic approach for the treatment of bone marrow failure syndromes such as FA, characterized by evident functional defects at the HSPC level.

## Statistical analysis

The statistical analysis was performed using GraphPad Prism version 5.0 for Windows. For the analyses of experiments in which $n < 5$, a nonparametric two-tailed Mann–Whitney test was performed when two variables were compared, or Kruskal–Wallis with Dunn's multiple comparison test when more than two variables were compared. In the experiments in which $n \geq 5$, a Kolmogorov–Smirnov test was done to test the normal distribution of the samples. If samples showed a normal distribution, a parametric two-tailed paired *t*-test was performed when two variables were compared or an ANOVA with Bonferroni's multiple comparisons test or Tukey's multiple comparisons test when more than two variables were compared. If samples did not follow normal distribution, the previously mentioned nonparametric tests were used. Significances are indicated in the figures and legends.

**Expanded View** for this article is available online.

## Acknowledgements

The authors would like to thank Aurora de la Cal for her assistance with the coordination in the delivery of the samples from the patients. Rebeca Sánchez-Domínguez for her expertise in flow cytometry and Centro de Transfusiones de la Comunidad de Madrid for providing cord blood samples. The authors are also indebted to the FA patients, families, and clinicians from the FA spanish network. This work was supported by grants from the "7th Framework Program European Commission (HEALTH-F5-2012-305421; EUROFANCOLEN)", "Ministerio de Sanidad, Servicios Sociales e Igualdad" (EC11/060 and EC11/550), "Ministerio de Economía, Comercio y Competitividad y Fondo Europeo de Desarrollo Regional (FEDER)" (SAF2015-68073-R), and "Fondo de Investigaciones Sanitarias, Instituto de Salud Carlos III" (RD12/0019/0023). The authors also thank the Fundación Marcelino Botín for promoting translational research at the Hematopoietic Innovative Therapies Division of the CIEMAT. CIBERER is an initiative of the Instituto de Salud Carlos III, Spain.

## Author contributions

BD, PG, AL, LN, JAB, PR: Conceived and designed the experiments. BD, PG, FJR-R, LA, GS, LU, SR-P, AL, LN, JAB, PR: Conducted the experiments. JS, CDH and MCH: Provided reagents, tools, samples from patients and contributed with ideas. BD, PG, LN, JAB, PR: Wrote the manuscript.

## Conflict of interest

M.C.H. is a current employee of Sangamo Therapeutics, Inc. The Division of Hematopoietic Innovative Therapies receives funding from Rocket Pharma. The rest of the authors declare no competing financial interests.

## For more information

Fanconi anemia research Foundation: www.anemiadefanconi.org and www.fanconi.org.

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
