## [Review Process File · EMBO Molecular Medicine]

Therapeutic Gene Editing in CD34⁺ Hematopoietic Progenitors from Fanconi Anemia Patients

Begoña Diez, Pietro Genovese, Francisco J. Roman-Rodriguez, Lara Alvarez, Giulia Schiroli, Laura Ugalde, Sandra Rodríguez-Perales, Julián Sevilla, Cristina Díaz de Heredia, Michael C. Holmes, Angelo Lombardo, Luigi Naldini, Juan A. Bueren and Paula Rio

Corresponding authors: Juan Bueren Paula Rio, CIEMAT/ CIBERER-ISC-III and Instituto de Investigacion Sanitaria Fundaci n Jimenez Diaz, IIS-FJD, UAM

Review timeline:	Submission date:	05 January 2017
	Editorial Decision:	10 March 2017
	Revision received:	10 July 2017
	Editorial Decision:	02 August 2017
	Revision received:	09 August 2017
	Accepted:	14 August 2017

Transaction Report:

Editor: Roberto Buccione

1st Editorial Decision

10 March 2017

Thank you for the submission of your manuscript to EMBO Molecular Medicine. We have now heard back from the Reviewers whom we asked to evaluate your manuscript.

I apologise again for the very significant delay in reaching a decision on your manuscript. In this case, we first experienced significant difficulties in securing expert and willing reviewers and obtaining their evaluations in a timely manner. In addition, we deemed one evaluation to be not sufficiently informative and I thus had to search for an additional one, thus causing further delay.

Hopefully the inevitable frustration due to this delay will be somewhat tempered by the generally positive evaluations.

As you will see the evaluations clearly recognize the relevance and clinical impact of they study, but also raise several important and to some degree overlapping concerns that encompass issues such as lack of controls, insufficient experimental detail or description of the cells used, improper statistical treatment, poor data presentation, and the need for further experimental to support validation. Perhaps reviewer #3 is less enthusiastic but I find that the well-taken points s/he raises (including concerns on data presentation quality) can be addressed.

In conclusion, while publication of the paper cannot be considered at this stage, we would be pleased to consider a revised submission, with the understanding that the Reviewers' concerns must be addressed with additional experimental data where appropriate and that acceptance of the

manuscript will entail a second round of review.

Please note that it is EMBO Molecular Medicine policy to allow a single round of revision only and that, therefore, acceptance or rejection of the manuscript will depend on the completeness of your responses included in the next, final version of the manuscript.

As you know, EMBO Molecular Medicine has a "scooping protection" policy, whereby similar findings that are published by others during review or revision are not a criterion for rejection. I understand that in this case, to address the above might entail a significant amount of additional work and time and might be technically challenging. However, I do ask you to get in touch with us after three months if you have not completed your revision, to update us on the status. Please also contact us as soon as possible if similar work is published elsewhere.

EMBO Molecular Medicine now requires a complete author checklist (<http://embomolmed.embopress.org/authorguide#editorial3>) to be submitted with all revised manuscripts. I am attaching a copy of the checklist to this letter for your convenience.

Please note that we now mandate that ALL corresponding authors list an ORCID digital identifier. You may do so through our web platform upon submission and the procedure takes <90 seconds to complete. We also encourage co-authors to supply an ORCID identifier, which will be linked to their name for unambiguous name identification.

Last, but not least, please carefully conform to our author guidelines (<http://embomolmed.embopress.org/authorguide>) to ensure rapid pre-acceptance processing in case of a favourable outcome on your revision.

I look forward to seeing a revised form of your manuscript as soon as possible.

***** Reviewer's comments *****

Referee #1 (Remarks):

Diez et al., MS # EMM-2017-07540
Therapeutic Gene Editing in CD34+ Hematopoietic Progenitors from Fanconi Anemia Patients

In this manuscript, the authors report on the successful directed targeting of reporter and therapeutic transgenes into the AAVS1 locus of healthy and, more importantly, Fanconi anemia patients' derived CD34+ cells. The development of safe and clinically compliant systems for gene targeting/editing is very much at the current attention of gene therapy research. Pioneering studies from the last 2-3 years have demonstrated the feasibility of modifying the genome of adult HSPC in a targeted manner, and the usefulness of doing so for monogenic hematological conditions such as SCID-X1 and CGD. Diez et al. take this approach one step further and apply it to the correction of FA-A patients' HSPC. As the authors convincingly argue, FA constitutes an especially interesting case study, inasmuch as i) HSPC numbers and quality are severely compromised in this disease; ii) targeted gene modification might be impaired in cells having a non-functional FA DNA repair pathway; and iii) FA patients would particularly benefit from a targeted gene correction strategy, even at low efficiency, on account of the well-known proliferation advantage of FA-corrected HSPC. Overall, the present study is justified, well conceived and conducted, and the results support the authors' conclusions. I do not have any major concerns with the current manuscript, but I list below some suggestions that I believe could improve it for publication:

1.- In page 8, paragraph 4, the authors state "Taken together, the in vivo experiments conducted with CB CD34+ cells that had been edited with a non-therapeutic donor demonstrated the efficient targeting of primitive human HSCs characterized by multipotent in vivo repopulating properties." Strictly speaking, secondary transplantation experiments would be necessary to draw that conclusion. Please, provide the results of such experiments, or tone down the conclusion accordingly.

2.- The authors make a compelling case on the targeted integration of EGFP or FANCA in healthy

cord blood-derived CD34+ cells, including transplantation assays and analysis of off-target sites. However, the experiments on mobilized peripheral blood CD34+ cells from FA patients appear somewhat rushed and much less well characterized. I understand that the amount of sample material available for these studies could be very limited, but showing actual transgene integration for more than one patient would be desirable. Also in this respect, the variability in phenotypic correction among the 5 patients analyzed is quite striking. Was it due to technical/methodological differences (amount of available cells, collection method, etc.), or to intrinsic differences in the patients' cells? Perhaps comparing with results from "conventional" LV-mediated correction in the same patients could be informative.

3.- In Fig. 1A, the origin of the variability in % of EGFP+ cells for some samples is unclear. The fact that statistical significance was not achieved appears more related to the high intra-sample experimental variation than to the median effect (which for some samples -e.g. FA-122, is quite apparent). Since these are LCLs, which should not be of limited availability, I would suggest repeating the experiment to clarify this issue.

4.- In page 5, paragraph 3, were the FA-A LCLs edited with PGK-FANCA/PuroR resistant to puromycin? The authors state later on in Discussion that puromycin selection was not routinely used (instead, they took advantage of the loss of mitomycin C sensitivity in these cells), but were they actually resistant?

5.- The authors explain the absence of EGFP signal in EGFP/PGK-FANCA-modified cells to the AAVS1 promoter not being active enough in human HSPC. To support this, they should show that their cassette is actually functional (and EGFP expressed) in other cell types.

6.- Some gel images (e.g. Fig. 1B, 5' PCR; Fig. 2 D, 5' and 3' PCRs) are below standard publication quality. Please, replace.

7.- I believe that readers would find it informative if the authors presented in the manuscript discussion a roadmap for the clinical implementation of the proposed strategies.

8.- Throughout the manuscript; please define abbreviations the first time they are used (e.g.: SCID, page 2; IDLV, page 5).

9.- Abstract, line 7, please remove extra comma.

Referee #2 (Comments on Novelty/Model System):

The manuscript describes a robust gene correction method for Faconi anemia that may be clinically tractable.

Referee #3 (Comments on Novelty/Model System):

i think there is a lot left to be done including characterizing the cells transducer for growth, etc-
Would have been better to start with murine model

Referee #3 (Remarks):

The paper from Deiz et al attempts to demonstrate in vivo editing of HSCs using novel Zn finger nucleases. Reasoning that FA should display in vivo selective advantage upon correction, FA as a model disease would seem to be the ideal for such correction.

This is an encouraging start but additional work is required. Also, such an idea has not borne out in repeated attempts to correct HSCs. In addition, one wonders why the group did not start with correction of murine HSCs with treatment of murine KO. In any case a more extensive analysis is required here.

Figure 1: in spite of the overall modest differences, the difference between mutant and corrected is alarming high in 2 mutant lines. P values should be provided to assure differences. KD in the HD

lines would be helpful as another point of comparison. On what basis is the PCR equivalent to GFP expression? How is this judgment made?

Figure 2: no attempt or mention is made of correction efficiency. This could be done by IF for FANCA even if the GFP was designed not to express. These would seem to be important controls. Also, the IF for D2 is of low quality; should include DAPI controls as well.

Figure 4: The point of the SGM3 mice should be explicitly mentioned in the text. This is not explained at any point in the paper. For clarity sake, a legend should be in the body of the figure so one can see the squares and circles meaning. No statistics are provided along with p values. A median value for the SGM3 mice is not provided. In order to ensure similar proliferative effects, the GFP+ and neg hCD45 cells should be sorted and analyzed separately for growth to ensure no changes due to insertion.

Figure 5: 5A-no stats are provided with error bars and p values. Why is there a marked increase in the FA-712 and FA-739 cells without Zn finger added?

Referee #4 (Remarks):

Using Zinc Finger Nuclease (ZFNs) and homology-directed targeted integration in to the safe harbor AAVS1 locus Diez et al have described their results in the current manuscript on stable complementation a functional Fanconi Anaemia (FANCA) cDNA in to FA-deficient CD34+ hematopoietic stem and progenitor cells (HSPCs). This study is a follow-up on their earlier work published in 2014 in EMBO Molecular Medicine (Rio et al 2014) on the feasibility of correcting the phenotype of a DNA repair deficiency syndrome using gene-targeting and cell reprogramming strategies. Further extending the initial work, Diez et al. now targeted mobilized peripheral blood CD34+ HSPCs from FA-A patients demonstrating integration of the FANCA construct in the AAVS1 locus and phenotypic correction of FA-A deficiency in gene edited cells. By targeting a non-therapeutic EGFP-reporter donor construct in the AAVS1 locus of lymphoblast cells (LCLs) derived from FA-A patients the authors first addressed if FA-A plays any role in HDR pathway (given that Fanconi Anemia genes are implicated in HDR) and have established that FA-A deficient cells are capable of undergoing homology directed repair. Further, by inserting therapeutic FANCA donors in FA-A patients LCLs, they showed a functional correction (MMC resistance and FANCD2 foci formation) in FANCA-complemented cells. Then, the authors targeted cord blood CD34+ HSPCs from healthy donors with a reporter construct and a therapeutic (FANCA) construct and report that the targeting efficiency was approximately 11%. Furthermore, they show that the gene targeted HSPCs were able to give rise to multi-lineage long-term reconstitution in the NSG and NSG-SGM3 mice. Most importantly, the authors have recapitulated the therapeutic gene complementation experiments in mobilized peripheral blood CD34+ HSPCs isolated from FA-A patients and showed that functional complementation of FANCA restores the resistance toward MMC treatment. Overall, the work is very impactful as it is performed in the most clinically relevant cell-type (HSPCs) for therapeutic gene complementation. The experiments are well designed and the interpretations and conclusions drawn are largely supported by the data, although some key issues need to be addressed.

Major Concerns

1. In the current version of the manuscript, its not mentioned if all four FA patient-derived LCLs (FA-55, FA-56, FA-122, FA-378) harbor the same mutations in FANCA gene. If the mutations are not the same, can the authors elaborate on the severity of each of the mutations - ie is there any residual activity associated with any of these mutations - for example could it be that the alleles associated with the FA-55 or FA-56 have residual activity whereas the alleles associated with FA-122 and FA-378 are more severe - with such differences potentially accounting for the varied rates of editing noted in each of the lines. Experiments quantifying the mitomycin C sensitivity of each of the lines should be performed to addresses these issues if information regarding mutation severity ius not already been established in these different lines.

2. Throughout the manuscript the AAVS1 targeted FANC-A lines are mentioned as FA-"corrected" which is misleading. There is no gene correction performed in these experiments rather a correct

copy of FANCA cDNA was knocked-in to the AAVS1 safe-harbor locus. Therefore, the author should use be careful; about their terminology as not to mislead - for example, rather "corrected" being used to describe the gene editing in the targeted cells, perhaps the term "complemented" or "knock-in" could be used. This advice should be heeded in the figures as well - for example Fig1A-B, it is misleading to refer to the targeted lines as "corrected" as thi9s implies HDR-mediated correction at the endogenous locus, which was not what was done in these experiments.

3. In the figure legend of Figure 2, the authors have mentioned that cells were nucleofected with the EGFP/PGK- FANCA donor IDLV (in left panel), or PGK-FANCA/PuroR donor IDLV (in right panel). However, in the result section, they have mentioned that FA-A LCLs edited with the EGFP/PGK-FANCA donor and the ZFNs mRNA reverted the characteristic mitomycin C (MMC) hypersensitivity of FA-A LCLs (Figure 2B), no EGFP expression was observed in these cells (data not shown). Similar results were obtained when cells were treated with the PGK- FANCA/PuroR (Figure 2B). Next, they show the results from "in and out" PCR in Figure 2D, showing correct insertion of donor template. Furthermore, in materials and methods section, it is mentioned that the cells were transduced with IDLV donors. If they have used the IDLV plasmid DNA vectors as donor and not the actual integration-defective lentiviruses that are correctly targeted in to AAVS1 then the interpretation drawn are based on FANCA expression from the nucleofected donor plasmid and not the actual integration of the donor in the AAVS1 locus. Can the authors clarify - has viral transduction or nucleofection been performed? In addition, standard deviation, number of experiments performed and statistical analysis is missing from Figure 2B. The quality of in and out PCR results is very poor (Figure 2D).

4. For phenotypic correction the authors have used MMC sensitivity and have shown that gene complemented cells are more resistant towards MMC treatment. Oxidative stress is implicated in the pathogenesis of FA and following functional complementation of FA-A, the oxidative burden should decrease in edited cells compared to the FA-A deficient cells. The authors could strengthen their interpretations to independently validate the phenotypic restoration using standard oxidative stress assays in FA_A LCL and/or in FA_A HSPCs.

Minor concerns:

1. In Figure 1 and 2, without ZFN controls are missing - though these controls have been provided in other experiments (eg. Those shown in EV2).

2. The authors show that immunophenotypically primitive cells post- 10 days culture have been gene edited and provide 3 month in vivo xenotransplantation data to support an interpretation that the most primitive HSCs have been targeted. As these systems are known in the field to be imperfect for determination of the presence of bona fide HSCs, the authors should soften their language to reflect this -- ie these data are "suggestive" gene targeting in HSCs.

3. Page 7, figure 4D is not referenced in the text, next to the results.

4. Page 9 "with a marked increase in MMC sensitivity" should be corrected to "with a marked decrease in MMC sensitivity or increased resistance towards MMC toxicity". Similarly Figure 5 legend caption "correction of MMC sensitivity" should be "restoration of resistance towards MMC toxicity".

1st Revision - authors' response

10 July 2017

Referee #1 (Remarks):

In this manuscript, the authors report on the successful directed targeting of reporter and therapeutic transgenes into the AAVS1 locus of healthy and, more importantly, Fanconi anemia patients' derived CD34+ cells. The development of safe and clinically compliant systems for gene targeting/editing is very much at the current attention of gene therapy research. Pioneering studies from the last 2-3 years have demonstrated the feasibility of modifying the genome of adult HSPC in a targeted manner, and the usefulness of doing so for monogenic hematological conditions such as SCID-X1 and CGD. Diez et al. take this approach one step further and apply it to the correction of

FA-A patients' HSPC. As the authors convincingly argue, FA constitutes an especially interesting case study, inasmuch as i) HSPC numbers and quality are severely compromised in this disease; ii) targeted gene modification might be impaired in cells having a non-functional FA DNA repair pathway; and iii) FA patients would particularly benefit from a targeted gene correction strategy, even at low efficiency, on account of the well-known proliferation advantage of FA-corrected HSPC. Overall, the present study is justified, well-conceived and conducted, and the results support the authors' conclusions. I do not have any major concerns with the current manuscript, but I list below some suggestions that I believe could improve it for publication:

We really appreciate Reviewer 1's comments. We have tried to answer the reviewer's suggestions one by one.

1.- In page 8, paragraph 4, the authors state "Taken together, the in vivo experiments conducted with CB CD34+ cells that had been edited with a non-therapeutic donor demonstrated the efficient targeting of primitive human HSCs characterized by multipotent in vivo repopulating properties." Strictly speaking, secondary transplantation experiments would be necessary to draw that conclusion. Please, provide the results of such experiments, or tone down the conclusion accordingly.

As proposed by the reviewer we have conducted secondary transplants in immunodeficient mice to demonstrate that long term HSCs have been targeted with our gene editing protocol. Results have been included as Supplementary Figure EV5 and commented in page 15 second paragraph and page 19 first paragraph. We have observed that the efficacy of gene editing in cells that engrafted secondary recipients was lower as compared to primary recipients. This observation is consistent with data from Genovese et al., 2014, so we have discussed this observation in the revised manuscript.

2.- The authors make a compelling case on the targeted integration of EGFP or FANCA in healthy cord blood-derived CD34+ cells, including transplantation assays and analysis of off-target sites. However, the experiments on mobilized peripheral blood CD34+ cells from FA patients appear somewhat rushed and much less well characterized. I understand that the amount of sample material available for these studies could be very limited, but showing actual transgene integration for more than one patient would be desirable. Also in this respect, the variability in phenotypic correction among the 5 patients analyzed is quite striking. Was it due to technical/methodological differences (amount of available cells, collection method, etc.), or to intrinsic differences in the patients' cells? Perhaps comparing with results from "conventional" LV-mediated correction in the same patients could be informative.

The reviewer raises a key point concerning the limited amount of CD34+ cells from FA patients to conduct these experiments. In spite of this difficulty we have now included the analysis of the specific integration in cells from patient FA-712, both in the absence and presence of MMC using a different pair of primers (Figure 5B, right panel). Why there is variability of the phenotypic correction in edited CD34+ cells from FA patients is difficult to answer. All possibilities proposed by the reviewer could certainly account for this. We have previously observed that even when working with LVs, differences in the correction of the MMC-hypersensitivity can be significant among patients and experiments.

3.- In Fig. 1A, the origin of the variability in % of EGFP+ cells for some samples is unclear. The fact that statistical significance was not achieved appears more related to the high intra-sample experimental variation than to the median effect (which for some samples -e.g. FA-122, is quite apparent). Since these are LCLs, which should not be of limited availability, I would suggest repeating the experiment to clarify this issue.

Following the reviewer's suggestion we have conducted two additional experiments to confirm our hypothesis. Once again the high variability in the proportion of EGFP+ cells among the different experiments did not allow us to define significant differences between edited and unedited FA LCLs, even though the tendency was still maintained.

4.- In page 5, paragraph 3, were the FA-A LCLs edited with PGK-FANCA/PuroR resistant to puromycin? The authors state later on in Discussion that puromycin selection was not routinely used

(instead, they took advantage of the loss of mitomycin C sensitivity in these cells), but were they actually resistant?

We have now mentioned in page 11 (third paragraph) that cells edited with PGK-FANCA/Puro^R donor and ZFNs are more resistant to puromycin than parental counterparts. However, we have observed that selection of edited FA cells is more efficient with MMC than with puromycin, probably because of the natural hypersensitivity of uncorrected FA cells to MMC.

5.- The authors explain the absence of EGFP signal in EGFP/PGK-FANCA-modified cells to the AAVS1 promoter not being active enough in human HSPC. To support this, they should show that their cassette is actually functional (and EGFP expressed) in other cell types.

In our previous paper (Rio *et al.*, EMM 2014) we detected EGFP expression in fibroblasts from FA-A patients using the same vector and AAVS1-ZFNs. Therefore, we have now mentioned this observation in second paragraph of page 18.

6.- Some gel images (e.g. Fig. 1B, 5' PCR; Fig. 2 D, 5' and 3' PCRs) are below standard publication quality. Please, replace.

We have improved the quality of Fig.1B, 5' PCR and substituted the image of Figure 2D with another one with improved quality. In this revised version we have also included in-out PCR results for PGK-FANCA/Puro^R donor IDLV in Figure 2E.

7.- I believe that readers would find it informative if the authors presented in the manuscript discussion a roadmap for the clinical implementation of the proposed strategies.

Thank you for this suggestion. We have now included a paragraph to explain this possibility in the discussion (last paragraph page 19).

8.- Throughout the manuscript; please define abbreviations the first time they are used (e.g.: SCID, page 2; IDLV, page 5).

We have now defined the different abbreviations the first time they are used in the manuscript.

9.- Abstract, line 7, please remove extra comma.

Thank you. We have now removed it.

Referee #2 (Comments on Novelty/Model System):

The manuscript describes a robust gene correction method for Fanconi anemia that may be clinically tractable.

We really appreciate the comment from the referee.

Referee #3 (Comments on Novelty/Model System):

I think there is a lot left to be done including characterizing the cells transducer for growth, etc-

We thank the reviewer for their careful reading of the manuscript and comments. We have now tried to answer their concerns and suggestions one by one. We have also included new experiments in the manuscript that we hope will clarify the reviewer's concerns.

Would have been better to start with murine model

We agree with the reviewer that the use of a murine model could be a good approach to demonstrate the feasibility of gene targeting in FA. In fact, other studies from the laboratory

have attempted to edit mouse FA HSPCs, although gene editing in mouse HSCs seems to be even more challenging than in human HSCs. This fact, together with the stronger proliferative advantage that we have observed in gene corrected human FA HSPCs versus mouse counterparts, encouraged us to conduct these experiments in human FA HSPCs. Additionally, and since gene editing tools have to be specifically designed to target human or mouse HSCs, we decided to focus directly in human HSCs for the future application of this approach in the clinics.

Referee #3 (Remarks):

The paper from Diez et al attempts to demonstrate *in vivo* editing of HSCs using novel Zn finger nucleases. Reasoning that FA should display *in vivo* selective advantage upon correction, FA as a model disease would seem to be the ideal for such correction. This is an encouraging start but additional work is required. Also, such an idea has not borne out in repeated attempts to correct HSCs. In addition, one wonders why the group did not start with correction of murine HSCs with treatment of murine KO. In any case a more extensive analysis is required here.

We thank the reviewer for their comments and we have tried to answer his/her concerns and suggestions one by one. We have also included new experiments in the manuscript that we hope can clarify the reviewer's concerns.

We agree with the reviewer that previous results have not been able to demonstrate proliferative advantage of HSCs from FA patients after gene correction. However, we have recently shown evident *in vivo* proliferative advantage of LV-mediated corrected HSPCs from FA-A patients after transplantation into immunodeficient mice (Rio and Navarro *et al.*, Blood under second revision). These FA HSPCs consisted of mPB CD34⁺ cells from FA-A patients treated with G-CSF and Plerixafor, which is exactly the cell type used in our current study.

Regarding the possibility of conducting these experiments in a FA KO model, this would also be an interesting approach. In fact, other studies from the laboratory have attempted to edit mouse FA HSPCs, although in these studies it was not possible to engraft edited cells in irradiated recipient mice, probably because of the high toxicity related to the gene editing of mouse HSCs. Additionally, and since gene editing tools have to be specifically designed to target human or mouse HSCs, we decided to focus directly in human HSCs for the future application of this approach in the clinics. The fact that the proliferation advantage of human FA HSPCs is much stronger as compared to FA mouse HSCs also encouraged us to conduct these experiments with human FA HSCs, the final target of our gene editing approach.

Figure 1: in spite of the overall modest differences, the difference between mutant and corrected is alarming high in 2 mutant lines. P values should be provided to assure differences. KD in the HD lines would be helpful as another point of comparison. On what basis is the PCR equivalent to GFP expression? How is this judgment made?

We have performed two new experiments to confirm differences between not complemented and FA-A LCLs complemented with LVs. However, statistical differences could not be observed, either when considering individual FA-LCLs (complemented and not complemented), or when considering the mean values of all these cell lines. Although the reviewer is perfectly right in their observation that a marked increase of EGFP expression seems to occur in complemented FA-122 and FA-378 LCLs, neither in these two cases statistical differences were observed. P values have been included both in the text, and also in Figure 1A.

Related to the equivalence of PCR analysis and EGFP expression, we have now initially limited our discussion to percentage of EGFP-positive cells. Only after confirmation that cells positive for EGFP expression correspond to gene edited cells (Figure 1B, 1C, 3C and 4A), we associate the determination of EGFP⁺ cells to edited cells.

Figure 2: no attempt or mention is made of correction efficiency. This could be done by IF for FANCA even if the GFP was designed not to express. These would seem to be important controls. Also, the IF for D2 is of low quality; should include DAPI controls as well.

To answer the question we have tried to isolate, without success, individual clones from edited LCLs. Estimating the actual efficacy of gene editing in FA cells is difficult since corrected clones develop proliferation advantage.

Following the reviewer's suggestion we have now modified and improved the quality of FANCD2 foci microphotographs including DAPI staining.

Figure 4: The point of the SGM3 mice should be explicitly mentioned in the text. This is not explained at any point in the paper. For clarity sake, a legend should be in the body of the figure so one can see the squares and circles meaning. No statistics are provided along with p values. A median value for the SGM3 mice is not provided. In order to ensure similar proliferative effects, the GFP+ and neg hCD45 cells should be sorted and analyzed separately for growth to ensure no changes due to insertion.

Trying to clarify the point raised by the reviewer, we have now included legends for Figure 4C, Figure EV4 and Figure EV5. Also a comment has been included in the text to discuss the relevance of the different immunodeficient mice that have been used (last paragraph, page 13). Median values of engraftment in NSG-SGM3 mice have also been included and p values for the different analyses conducted in NSG and NSG-SGM3 mice have been included in Figure 4C, D, E and F, Figure EV4 and Figure EV5.

Figure 5: 5A-no stats are provided with error bars and p values. Why is there a marked increase in the FA-712 and FA-739 cells without Zn finger added?

Each of these figures corresponds to individual patients. Additionally, due to the very limited number of mPB CD34⁺ cells and samples from the FA patients, we could not conduct three different experiments with these cells. For this reason, error bars and p values could not be included for each patient. However, using MMC-resistant data from all these patients, we have determined mean values of MMC resistance corresponding to edited cells versus those transduced only with donor vectors. P values demonstrate the significant increase resistance to MMC in edited cells.

Regarding the increase in MMC survival in FA-712 and FA739 in the absence of ZFNs, we think this can be due to higher toxicity of these particular cells after transfections in the presence of ZFNs, in contrast to transfections in the absence of ZFNs.

Referee #4 (Remarks):

Using Zinc Finger Nuclease (ZFNs) and homology-directed targeted integration in to the safe harbor AAVS1 locus Diez et al have described their results in the current manuscript on stable complementation a functional Fanconi Anaemia (FANCA) cDNA in to FA-deficient CD34⁺ hematopoietic stem and progenitor cells (HSPCs). This study is a follow-up on their earlier work published in 2014 in EMBO Molecular Medicine (Rio et al 2014) on the feasibility of correcting the phenotype of a DNA repair deficiency syndrome using gene-targeting and cell reprogramming strategies. Further extending the initial work, Diez et al. now targeted mobilized peripheral blood CD34⁺ HSPCs from FA-A patients demonstrating integration of the FANCA construct in the AAVS1 locus and phenotypic correction of FA-A deficiency in gene edited cells. By targeting a non-therapeutic EGFP-reporter donor construct in the AAVS1 locus of lymphoblast cells (LCLs) derived from FA-A patients the authors first addressed if FA-A plays any role in HDR pathway (given that Fanconi Anemia genes are implicated in HDR) and have established that FA-A deficient cells are capable of undergoing homology directed repair. Further, by inserting therapeutic FANCA donors in FA-A patients LCLs, they showed a functional correction (MMC resistance and FANCD2 foci formation) in FANCA-complemented cells. Then, the authors targeted cord blood CD34⁺ HSPCs from healthy donors with a reporter construct and a therapeutic (FANCA) construct and report that the targeting efficiency was approximately 11%. Furthermore, they show that the gene

targeted HSPCs were able to give rise to multi-lineage long-term reconstitution in the NSG and NSG-SGM3 mice. Most importantly, the authors have recapitulated the therapeutic gene complementation experiments in mobilized peripheral blood CD34+ HSPCs isolated from FA-A patients and showed that functional complementation of FANCA restores the resistance toward MMC treatment. Overall, the work is very impactful as it is performed in the most clinically relevant cell-type (HSPCs) for therapeutic gene complementation. The experiments are well designed and the interpretations and conclusions drawn are largely supported by the data, although some key issues need to be addressed.

We are grateful to the reviewer for the nice recap of our study and their suggestions to improve the quality of the paper. We have tried to answer reviewer concerns one by one.

Major Concerns

1. In the current version of the manuscript, it's not mentioned if all four FA patient-derived LCLs (FA-55, FA-56, FA-122, FA-378) harbor the same mutations in FANCA gene. If the mutations are not the same, can the authors elaborate on the severity of each of the mutations - ie is there any residual activity associated with any of these mutations - for example could it be that the alleles associated with the FA-55 or FA-56 have residual activity whereas the alleles associated with FA-122 and FA-378 are more severe - with such differences potentially accounting for the varied rates of editing noted in each of the lines. Experiments quantifying the mitomycin C sensitivity of each of the lines should be performed to address these issues if information regarding mutation severity is not already been established in these different lines.

We would like to thank the reviewer for their suggestion since we agree that this was not clear in the first version of the manuscript. In this revised version we have now included (in Table EV1) the description of the mutations in the different FA-A LCLs used. However, we think that the different mutations of these patients do not account for differences in the HR efficiency between FA-A LCLs complemented with LV or not. In fact, although FA-55 and FA-122 patients carry the same mutation, the efficiency of gene editing in both LCLs, either in the presence or absence of a functional FANCA protein, is very different.

2. Throughout the manuscript the AAVS1 targeted FANCA lines are mentioned as FA-"corrected" which is misleading. There is no gene correction performed in these experiments rather a correct copy of FANCA cDNA was knocked-in to the AAVS1 safe-harbor locus. Therefore, the author should use be careful; about their terminology as not to mislead - for example, rather "corrected" being used to describe the gene editing in the targeted cells, perhaps the term "complemented" or "knock-in" could be used. This advice should be heeded in the figures as well - for example Fig1A-B, it is misleading to refer to the targeted lines as "corrected" as this implies HDR-mediated correction at the endogenous locus, which was not what was done in these experiments.

We agree with the reviewer that we have not corrected the genetic defect of FA cells. Therefore, when we talk about correction, we have specified that we are talking about correction of the phenotype. As suggested by the reviewer, in the case of the genetic defect of FA cells, we have substituted "corrected" by "complemented by gene editing".

3. In the figure legend of Figure 2, the authors have mentioned that cells were nucleofected with the EGFP/PGK- FANCA donor IDLV (in left panel), or PGK-FANCA/PuroR donor IDLV (in right panel). However, in the result section, they have mentioned that FA-A LCLs edited with the EGFP/PGK-FANCA donor and the ZFNs mRNA reverted the characteristic mitomycin C (MMC) hypersensitivity of FA-A LCLs (Figure 2B), no EGFP expression was observed in these cells (data not shown). Similar results were obtained when cells were treated with the PGK- FANCA/PuroR (Figure 2B). Next, they show the results from "in and out" PCR in Figure 2D, showing correct insertion of donor template. Furthermore, in materials and methods section, it is mentioned that the cells were transduced with IDLV donors. If they have used the IDLV plasmid DNA vectors as donor and not the actual integration-defective lentiviruses that are correctly targeted in to AAVS1 then the interpretation drawn are based on FANCA expression from the nucleofected donor plasmid and not the actual integration of the donor in the AAVS1 locus. Can the authors clarify - has viral transduction or nucleofection been performed? In addition, standard deviation, number of

experiments performed and statistical analysis is missing from Figure 2B. The quality of in and out PCR results is very poor (Figure 2D).

We apologize for the mistake in the Figure 2 legend. Actually, the FA-56 LCL was transduced with PGK-FANCA/Puro^R donor IDLV (in left panel), or EGFP/PGK-FANCA donor IDLV (in right panel) and afterwards nucleofected or not with AAVS1-ZFNs. Therefore, data in Figure 2 combines the use of IDLVs with AAVS1-ZFNs as mRNA. No plasmid DNA has been used in any of the experiments conducted in this manuscript. We have modified the Figure 2 legend to clarify this point.

Mean values of three different experiments have now been included in Figure 2B and statistical analysis has been conducted as the reviewer suggested.

Regarding the quality of Figure 2D, we have now replaced this figure with Figures 2D and 2E with new "in and out" PCR results corresponding to the 5' and 3' junctions of cells edited with the EGFP/PGK-FANCA (Figure 2E) and with the PGK-FANCA/Puro^R (Figure 2D) donor IDLVs. In all instances experiments were conducted in the absence and the presence of AAVS1-ZFNs. As the reviewer will see, the quality of these figures has been improved with respect to the previous ones.

4. For phenotypic correction the authors have used MMC sensitivity and have shown that gene complemented cells are more resistant towards MMC treatment. Oxidative stress is implicated in the pathogenesis of FA and following functional complementation of FA-A, the oxidative burden should decrease in edited cells compared to the FA-A deficient cells. The authors could strengthen their interpretations to independently validate the phenotypic restoration using standard oxidative stress assays in FA_A LCL and/or in FA_A HSPCs.

We are grateful to the reviewer for their suggestion. We have conducted analysis of ROS production in a LCL from a FA patient after gene editing. In this context, edited cells showed a reduction in ROS production in comparison with either FANCA-deficient counterparts or FA-A cells targeted only with the PGK-FANCA/Puro^R donor IDLV. As proposed by the reviewer this confirms the phenotypic correction of the FA pathway by another criterion. Results have been included in Figure EV1, described in the Results section (third paragraph, page 11) and highlighted in the second paragraph of page 18.

Minor concerns:

1. In Figure 1 and 2, without ZFN controls are missing - though these controls have been provided in other experiments (eg. Those shown in EV2).

As proposed by the reviewer, we have now included these controls in both figures.

2. The authors show that immunophenotypically primitive cells post- 10 days culture have been gene edited and provide 3 month in vivo xenotransplantation data to support an interpretation that the most primitive HSCs have been targeted. As these systems are known in the field to be imperfect for determination of the presence of bona fide HSCs, the authors should soften their language to reflect these data are "suggestive" gene targeting in HSCs.

We agree with the reviewer. Therefore, we have conducted secondary transplants with BM cells from primary immunodeficient mice to demonstrate that long term HSCs have been targeted with our gene editing protocol. Although the efficacy of gene editing was lower in cells that repopulated the hematopoiesis of secondary recipients, gene editing could be confirmed in these very primitive HSCs. Results have been included as Supplementary Figure EV5 and commented in page 15 second paragraph and page 19 first paragraph.

3. Page 7, figure 4D is not referenced in the text, next to the results

We have now mentioned the figure next to the results (first paragraph page 14).

4. Page 9 "with a marked increase in MMC sensitivity" should be corrected to "with a marked decrease in MMC sensitivity or increased resistance towards MMC toxicity". Similarly Figure 5

legend caption "correction of MMC sensitivity" should be "restoration of resistance towards MMC toxicity".

The reviewer is perfectly right. We have corrected both sentences as proposed by the reviewer.

2nd Editorial Decision

02 August 2017

Thank you for the submission of your revised manuscript to EMBO Molecular Medicine. We have now received the enclosed reports from the reviewers that were asked to re-assess it. As you will see the reviewers are now supportive, although reviewer 2 has a few final requests that require your action. Please also note that I asked reviewer 1 to also evaluate your responses to reviewer 4 as the latter was not available.

I am thus prepared to accept your manuscript for publication pending satisfactory compliance with reviewer 2's final requests and the following editorial requirements:

- 1) Please upload separate manuscript (word doc) and figure files (one per figure) including EV figures.
- 2) A manuscript callout appears to be missing for Table EV1
- 3) We are now encouraging the publication of source data, particularly for electrophoretic gels and blots, with the aim of making primary data more accessible and transparent to the reader. Would you be willing to provide a PDF file per figure that contains the original, uncropped and unprocessed scans of all or at least the key gels used in the manuscript? The PDF files should be labeled with the appropriate figure/panel number, and should have molecular weight markers; further annotation may be useful but is not essential. The PDF files will be published online with the article as supplementary "Source Data" files. If you have any questions regarding this just contact me.
- 4) Please format references as to show 20 authors et al. (currently 10 authors et al.).
- 5) Please provide 5 keywords in the title page.
- 6) Please include the information on age and gender of mice in the main text.

Please submit your revised manuscript within two weeks. I look forward to seeing a revised form of your manuscript as soon as possible.

***** Reviewer's comments *****

Referee #1 (Remarks):

Diez et al., MS # EMM-2017-07540-V2
Therapeutic Gene Editing in CD34+ Hematopoietic Progenitors from Fanconi Anemia Patients

In the revised version of their manuscript, the authors satisfactorily address all the points raised by Reviewer #1.

In addition to this, the authors also address all the key points of Reviewer #4 in a satisfactory manner (some of them shared with Reviewer #1), and have edited the manuscript text as per the reviewer's suggestions. Specifically, the authors now present in the revised version of their manuscript detailed information as to the mutations present in each patient, thus unlinking mutation type and edition efficiency; clarify the different transduction/nucleofection regimes in the figure legends and include necessary statistical information; replace images of gels that were of low quality; and include new experimental evidence on i) phenotypic correction as judged by reduced ROS production, and ii) performing secondary transplants to demonstrate targeting of long-term HSCs. Moreover, the authors now use 'gene complementation' instead of 'gene correction', and have edited the manuscript text to correct some mistakes and improve clarity as per the reviewers' suggestions.

Overall, I appreciate the additional effort the authors have put to including new experimental evidence and editing the manuscript text in response to the comments of Reviewers #1 and #4 and, at this point, I emphatically support publication of the revised version of this manuscript in EMBO Mol Med.

Referee #3 (Comments on Novelty/Model System):

There are missing error bars, p values, and poorly displayed data

Referee #3 (Remarks):

This is a revised paper on the Zn finger mediated correction of FA-A mutant cells. This is an interesting paper and the authors have made some attempts to address the previous reviews.

Fig 2: D2 foci should be shown as: 1-D2 foci 2-Dapi and 3-merged

Fig 4: the dashed lines and solid lines in the graphs should be labeled on the figure as to what mouse they represent.

Fig 5: it is still problematic that there are not replicates

2nd Revision - authors' response

09 August 2017

Point-by-point response

1) Please upload separate manuscript (word doc) and figure files (one per figure) including EV figures.

We have included separate docs for manuscript and individual figures.

2) A manuscript callout appears to be missing for Table EV1.

Thank you. We have included the reference to this Table at the end of the second paragraph in page 4.

3) We are now encouraging the publication of source data, particularly for electrophoretic gels and blots, with the aim of making primary data more accessible and transparent to the reader. Would you be willing to provide a PDF file per figure that contains the original, uncropped and unprocessed scans of all or at least the key gels used in the manuscript? The PDF files should be labeled with the appropriate figure/panel number, and should have molecular weight markers; further annotation may be useful but is not essential. The PDF files will be published online with the article as supplementary "Source Data" files. If you have any questions regarding this just contact me.

As proposed, we have now uploaded unprocessed pdf files corresponding to each of the gels presented in the manuscript.

4) Please format references as to show 20 authors et al. (currently 10 authors et al.).

We have modified the format of the references to include the first 20 authors.

5) Please provide 5 keywords in the title page.

We have included these five keywords in the title page.

6) Please include the information on age and gender of mice in the main text.

We have included this information in Material and Methods (second paragraph, page 8).

Referee #1 (Remarks):

**Diez et al., MS # EMM-2017-07540-V2
Therapeutic Gene Editing in CD34+ Hematopoietic Progenitors from Fanconi Anemia Patients**

In the revised version of their manuscript, the authors satisfactorily address all the points raised by Reviewer #1.

In addition to this, the authors also address all the key points of Reviewer #4 in a satisfactory manner (some of them shared with Reviewer #1), and have edited the manuscript text as per the reviewer's suggestions. Specifically, the authors now present in the revised version of their manuscript detailed information as to the mutations present in each patient, thus unlinking mutation type and edition efficiency; clarify the different transduction/nucleofection regimes in the figure legends and include necessary statistical information; replace images of gels that were of low quality; and include new experimental evidence on i) phenotypic correction as judged by reduced ROS production, and ii) performing secondary transplants to demonstrate targeting of long-term HSCs. Moreover, the authors now use 'gene complementation' instead of 'gene correction', and have edited the manuscript text to correct some mistakes and improve clarity as per the reviewers' suggestions.

Overall, I appreciate the additional effort the authors have put to including new experimental evidence and editing the manuscript text in response to the comments of Reviewers #1 and #4 and, at this point, I emphatically support publication of the revised version of this manuscript in EMBO Mol Med.

We really thank the reviewer for his/her positive opinion of the revised manuscript and also for reviewing our answers to reviewer 4.

Referee #3 (Comments on Novelty/Model System):

There are missing error bars, p values, and poorly displayed data.

As the reviewer could observe in figures 1, 2, 3 and 4, we have shown error bars and statistical analyses. Only in the case of figure 5 this was not possible because we could not perform replicate cultures from each patient, due to the scarcity in the number of cells: mPB CD34⁺ cells from FA-A patients. Even in this case, we have included MMC resistance data corresponding to treatments in the presence/absence of ZFNs in mPB CD34⁺ cells from 5 different patients. Therefore, mean values and standard error values, and statistical significance of differences have been included in the third paragraph of page 16.

Referee #3 (Remarks):

This is a revised paper on the Zn finger mediated correction of FA-A mutant cells. This is an interesting paper and the authors have made some attempts to address the previous reviews.

We thank the reviewer for his/her suggestions and the careful revision of the manuscript.

Fig 2: D2 foci should be shown as: 1-D2 foci 2-Dapi and 3-merged.

We have modified the figure as suggested by the reviewer.

Fig 4: the dashed lines and solid lines in the graphs should be labeled on the figure as to what mouse they represent.

Dashed lines and solid lines have been included in the legend of Figure 4, Figure EV4 and Figure EV5.

Fig 5: it is still problematic that there are not replicates.

In these experiments it was not possible to perform replicate cultures from each patient due to the scarcity in the number of cells: mPB CD34⁺ cells from FA-A patients. However, we have included MMC resistance data corresponding to treatments in the presence/absence of ZFNs in mPB CD34⁺ cells from 5 different patients. Consequently, mean values and standard error values, and statistical significance of differences have been included in the third paragraph of page 16.

Corresponding Author Name: Paula Rio

Manuscript Number: